# Limited inhibition of multiple nodes in a driver network blocks metastasis

**Ali Ekrem Yesilkanal[1], Dongbo Yang[1], Andrea Valdespino[1], Payal Tiwari[1], Alan U Sabino[2], Long Chi Nguyen[1], Jiyoung Lee[1], Xiao-He Xie[1], Siqi Sun[1], Christopher Dann[1], Lydia Robinson-Mailman[1], Ethan Steinberg[1], Timothy Stuhlmiller[3], Casey Frankenberger[1], Elizabeth Goldsmith[4], Gary L Johnson[3], Alexandre F Ramos[2]\*, Marsha R Rosner[1]\***

[1]Ben May Department for Cancer Research, University of Chicago, Chicago, United States; [2]Instituto do Câncer do Estado de São Paulo, Faculdade de Medicina and Escola de Artes, Ciências e Humanidades; University of São Paulo, São Paulo, Brazil; [3]Department of Pharmacology, University of North Carolina at Chapel Hill, Chapel Hill, United States; [4]UT Southwestern, Dallas, United States

**Abstract** Metastasis suppression by high-dose, multi-drug targeting is unsuccessful due to network heterogeneity and compensatory network activation. Here, we show that targeting driver network signaling capacity by limited inhibition of core pathways is a more effective anti-metastatic strategy. This principle underlies the action of a physiological metastasis suppressor, Raf Kinase Inhibitory Protein (RKIP), that moderately decreases stress-regulated MAP kinase network activity, reducing output to transcription factors such as pro-metastatic BACH1 and motility-related target genes. We developed a low-dose four-drug mimic that blocks metastatic colonization in mouse breast cancer models and increases survival. Experiments and network flow modeling show limited inhibition of multiple pathways is required to overcome variation in MAPK network topology and suppress signaling output across heterogeneous tumor cells. Restricting inhibition of individual kinases dissipates surplus signal, preventing threshold activation of compensatory kinase networks. This low-dose multi-drug approach to decrease signaling capacity of driver networks represents a transformative, clinically relevant strategy for anti-metastatic treatment.

**\*For correspondence:**
alex.ramos@usp.br (AFR);
m-rosner@uchicago.edu (MRR)

## Introduction

Cancer is a complex disease marked by heterogeneity. For solid tumors, metastatic dissemination of sub-populations of tumor cells throughout the body is primarily responsible for lethality (*Weigelt et al., 2005*). Metastasis is characterized by many distinct biological processes such as tumor cell invasion, transport in vessels, and colonization at distant sites that involve significant cellular stress. Metastatic progression is further complicated by dynamic changes in tumors that undergo evolutionary change in response to cells and stresses within the microenvironment.

Previous approaches to treating metastatic disease have largely been ineffective at preventing resistance or recurrence due to cellular heterogeneity and robust compensatory mechanisms. Targeting individual metastatic pathways at maximum tolerated doses using single or multiple anticancer agents can activate compensatory pathways that eventually overcome treatment (*Gallaher et al., 2018*; *Duncan et al., 2012*; *Wong et al., 2019*). Even high dose combination therapies that target single kinases across multiple networks can be toxic and are insufficient to enable long-term survival (*Westin et al., 2019*; *Liu et al., 2019*; *Robert et al., 2019*). The commonality between signaling pathways in both normal and tumor cells has also limited the efficacy of most therapeutic strategies. Therefore, novel strategies for suppressing metastasis are needed, particularly for cancers such as triple negative breast cancer (ER[-], PR[-], HER2[low]; TNBC) that lack effective targeted therapy.

An alternative approach to suppressing metastasis employs a phenomenological framework built upon understanding the action of physiological metastasis regulators. Biological metastasis suppressors are proteins that inhibit various steps of metastasis and are lost or silenced in metastatic tumors (*Zhao et al., 2015*). To date, approximately 100 metastasis suppressors have been identified, many of which also inhibit tumor growth (*Zhao et al., 2015*). Interestingly, several of these metastasis suppressors are kinases or proteins that modify signaling cascades and provide insight into metastatic signaling mechanisms. One of these, Raf Kinase Inhibitory Protein (RKIP; PEBP1), is a regulator of Raf kinase activity that is deleted or lost in virtually all metastatic solid tumors (reviewed by *Yesilkanal and Rosner, 2018*). Reintroducing RKIP to metastatic TNBC cells blocks invasion of cells in vitro and inhibits intravasation and metastasis of tumor cells in vivo (*Dangi-Garimella et al., 2009*). Numerous studies have shown that loss of RKIP protein expression is associated with poor outcome in a variety of tumors including breast, prostate, and melanoma (*Lamiman et al., 2014*). Furthermore, expression of RKIP in preclinical models enhances the response to chemotherapy as well as radiation suggesting that RKIP can potentiate therapeutic efficacy (*Bonavida et al., 2008*; *Chatterjee et al., 2004*; *Woods Ignatoski et al., 2008*). Therefore, RKIP provides a powerful model system for developing new anti-metastatic therapies based on the mechanism by which RKIP modulates signaling network dynamics and prevents metastatic transformation.

In the present study, we utilized the action of RKIP as a conceptual framework for a new strategy to target metastasis. Metastatic suppression is achieved by restricting but not eliminating the activities of multiple kinases in a driver signaling network, stress MAP kinases (MAPKs). One molecular output of the network, the transcription factor BACH1, drives clinically relevant metastatic motility genes. We developed a four-drug mimic of RKIP used at sub-therapeutic doses that inhibits network output, reduces metastasis, and improves survival in vivo. Modeling of different MAPK network topologies provides a rationale for this multi-drug anti-metastatic strategy that reduces signaling flow at multiple rather than single nodes and prevents activation of compensatory signaling pathways. These findings challenge the current approaches to drug treatment and suggest an alternative strategy for controlling metastatic disease in breast and potentially other cancers.

## Results

### RKIP regulates a clinically relevant set of motility-related genes driven by the pro-metastatic transcription factor BACH1

To characterize the mechanism by which RKIP suppresses metastasis, we first analyzed gene expression data from breast cancer patient samples in The Cancer Genome Atlas (TCGA) study. Our analysis revealed that RKIP (*PEBP1*) expression negatively correlated with genes involved in cell motility (cell leading edge, cell migration, focal adhesion) and kinase-mediated signaling (regulation of GTPase activity, phosphotransferase activity) (*Figure 1A*). Among the genes most inversely correlated with RKIP was *BACH1* (BTB and CNC homology 1), a pro-metastatic, basic leucine zipper transcription factor that is post-translationally inhibited by RKIP via *let-7* (*Figure 1A*; *Dangi-Garimella et al., 2009*; *Yun et al., 2011*). To test direct regulation of these motility genes by RKIP, we performed RNA sequencing (RNA-seq) of transcripts in control versus RKIP-expressing xenograft tumors of BM1, a bone-tropic, Ras/B-Raf mutant TNBC cell line derived from MDA-MB-231 (*Figure 1—figure supplement 1A,B*; *Kang et al., 2003*). Over 70 of the motility genes as well as *BACH1* that inversely correlate with RKIP in patients were also downregulated by RKIP in xenograft tumors, suggesting these genes are transcriptionally regulated by RKIP in TNBC (*Figure 1B,C* and *Figure 1—figure supplement 1C*). We validated the differential expression of 15 motility genes previously implicated in metastasis (*Qin et al., 2014*; *Flockhart et al., 2009*; *Yoeli-Lerner et al., 2005*; *Amano et al., 2010*; *Gadea and Blangy, 2014*; *Riento and Ridley, 2003*; *Zhang et al., 2017*; *Yu et al., 2015*; *Kobayashi et al., 2014*) using BM1 xenograft tumors expressing a more robust version of RKIP (S153E mutant) in an independent in vivo study (*Figure 1—figure supplement 1D*; *Dangi-Garimella et al., 2009*).

The RNA-seq analysis also revealed upregulation of genes related to mitochondrial metabolism and oxidative phosphorylation in RKIP-expressing BM1 tumors (*Figure 1—figure supplement 1C*). Moreover, the mitochondrial gene targets of RKIP positively correlated with RKIP, while negatively correlating with *BACH1* gene expression in breast cancer patients (*Figure 1—figure supplement 1E*

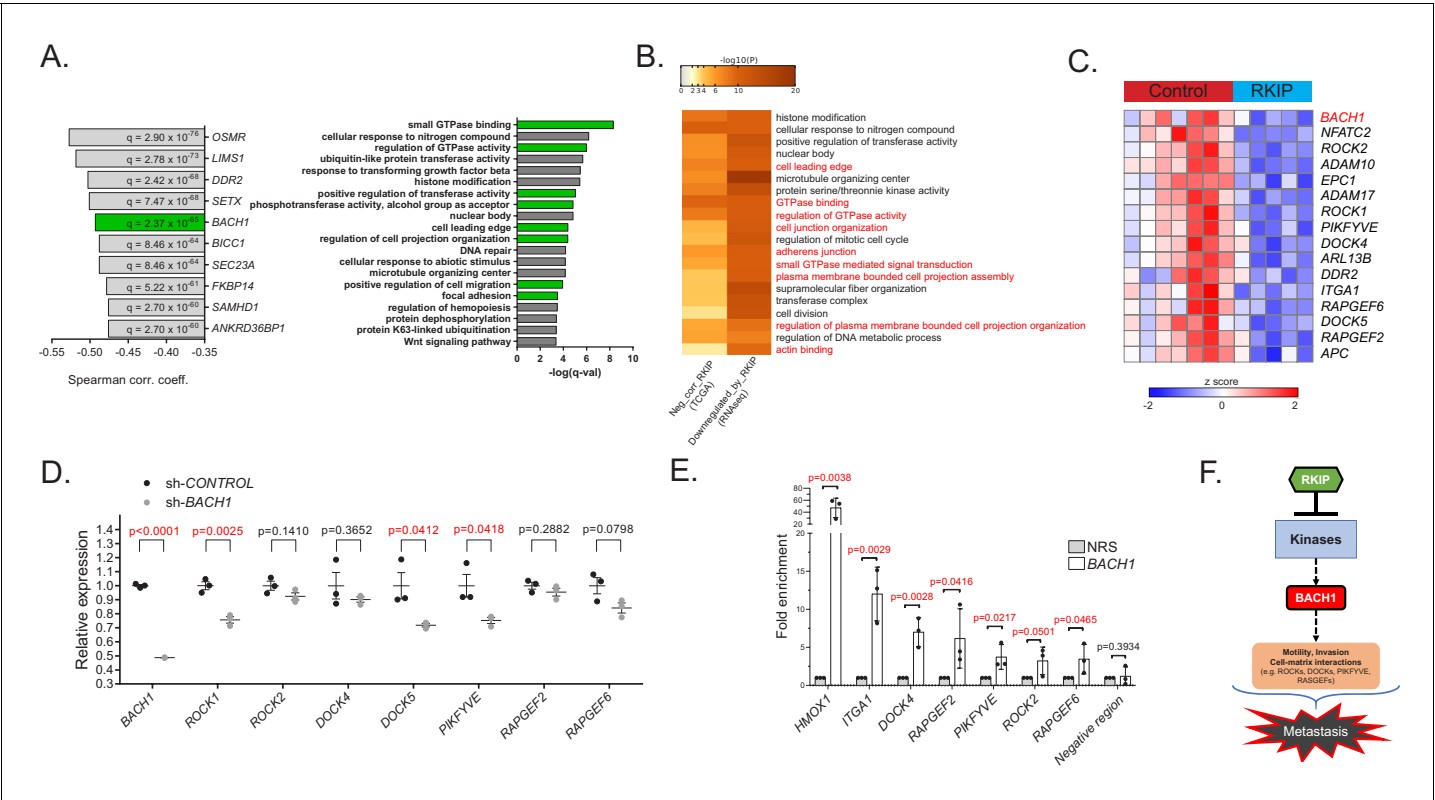

**Figure 1.** RKIP regulates a clinically relevant set of motility-related genes driven by the pro-metastatic transcription factor BACH1. (**A**) Left panel: Top 10 genes negatively correlated with RKIP (*PEBP1*) gene expression in TCGA BRCA samples (provisional, n = 1100), ranked by Spearman correlation coefficient. Right panel: Gene sets enriched in genes negatively correlated with RKIP in TCGA BRCA set. (**B**) Gene sets commonly enriched in genes negatively correlated with RKIP in TCGA BRCA set and genes downregulated by RKIP in the RNA-seq study. (**C**) A set of differentially expressed motility genes and *BACH1* gene expression in control (n = 7) vs. RKIP-expressing (n = 5) BM1 tumors. (**D**) qRT-PCR analysis of control (n = 3) and sh*BACH1*-expressing (n = 3) BM1 tumors, demonstrating downregulation of motility gene expression when BACH1 levels are reduced. Student's t-test, two-tailed. (**E**) Chromatin immuno-precipitation analysis of BACH1 binding in the promoter regions of the motility genes in BM1 cells. Mean ± s.e.m of three independent experiments. Student's t-test, one-tailed. NRS, normal rabbit serum (**F**) Summary diagram showing regulation of *BACH1* and motility gene expression by RKIP. For the source data, see *Figure 1—source data 1*.

The online version of this article includes the following source data and figure supplement(s) for figure 1:

**Source data 1.** Source data files for *Figure 1D and E*, *Figure 1—figure supplement 1D* and *Figure 1—figure supplement 2D*.

**Figure supplement 1.** Transcriptional regulation of motility-related genes by RKIP.

**Figure supplement 2.** Transcription of metastasis-related RKIP target genes is mediated in part by BACH1.

and *Figure 1—figure supplement 2A*). Our previous work similarly showed that reducing BACH1 expression in TNBC increased oxidative phosphorylation in mitochondria, mirroring the RKIP pheno-type (*Lee et al., 2019*). This prompted us to investigate whether BACH1 is in part responsible for regulating motility-related gene targets of RKIP as well. Indeed, *BACH1* gene expression positively correlates with RKIP-inhibited motility genes in both patient samples and xenograft TNBC tumors (*Figure 1—figure supplement 2A*). ENCODE and ChIP-seq analysis in BM1 cells shows BACH1 binding to the promoter regions of several motility genes (*Figure 1—figure supplement 2B,C*; *Landt et al., 2012*). To confirm that BACH1 transcriptionally regulates metastatic motility-related genes in TNBC cells, we performed qRT-PCR using BM1 xenograft tumors expressing shRNAs against *BACH1* in two independent mouse experiments (*Figure 1D*, *Figure 1—figure supplement 2D*, *Figure 1—source data 1*) and ChIP assays in BM1 cells to demonstrate direct BACH1 binding to motility gene promoters (*Figure 1E*, *Figure 1—source data 1*). These findings establish BACH1-controlled motility genes as pro-metastatic targets of RKIP and illustrate the clinical relevance of this regulatory system to TNBC patients (*Figure 1F*).

## RKIP targets multiple kinases in the stress MAPK network

RKIP inhibits the activity of Raf and other kinases in cultured cells (reviewed in *Yesilkanal and Rosner, 2018*). To identify kinases targeted by RKIP in TNBC tumors, we analyzed changes in kinase expression and activity by MIB-MS analysis (*Duncan et al., 2012*). To capture and quantify functional kinases in tumors, tumor lysates were exposed to kinase inhibitors covalently linked to Sepharose beads (MIBs) followed by mass spectrometry. Of the 248 captured kinases that were present in both control and RKIP-expressing BM1 tumors from mouse xenografts, RKIP significantly altered the functional capture of 30 kinases (*Figure 2A*, *Figure 2—source data 1*). Consistent with its role as a kinase suppressor, RKIP inhibited most of these kinases (23) including the previously identified RKIP target ERK2 (*Yeung et al., 1999*). The kinases targeted by RKIP were distributed across multiple branches of the kinome tree and not limited to a specific kinase class (*Figure 2B*).

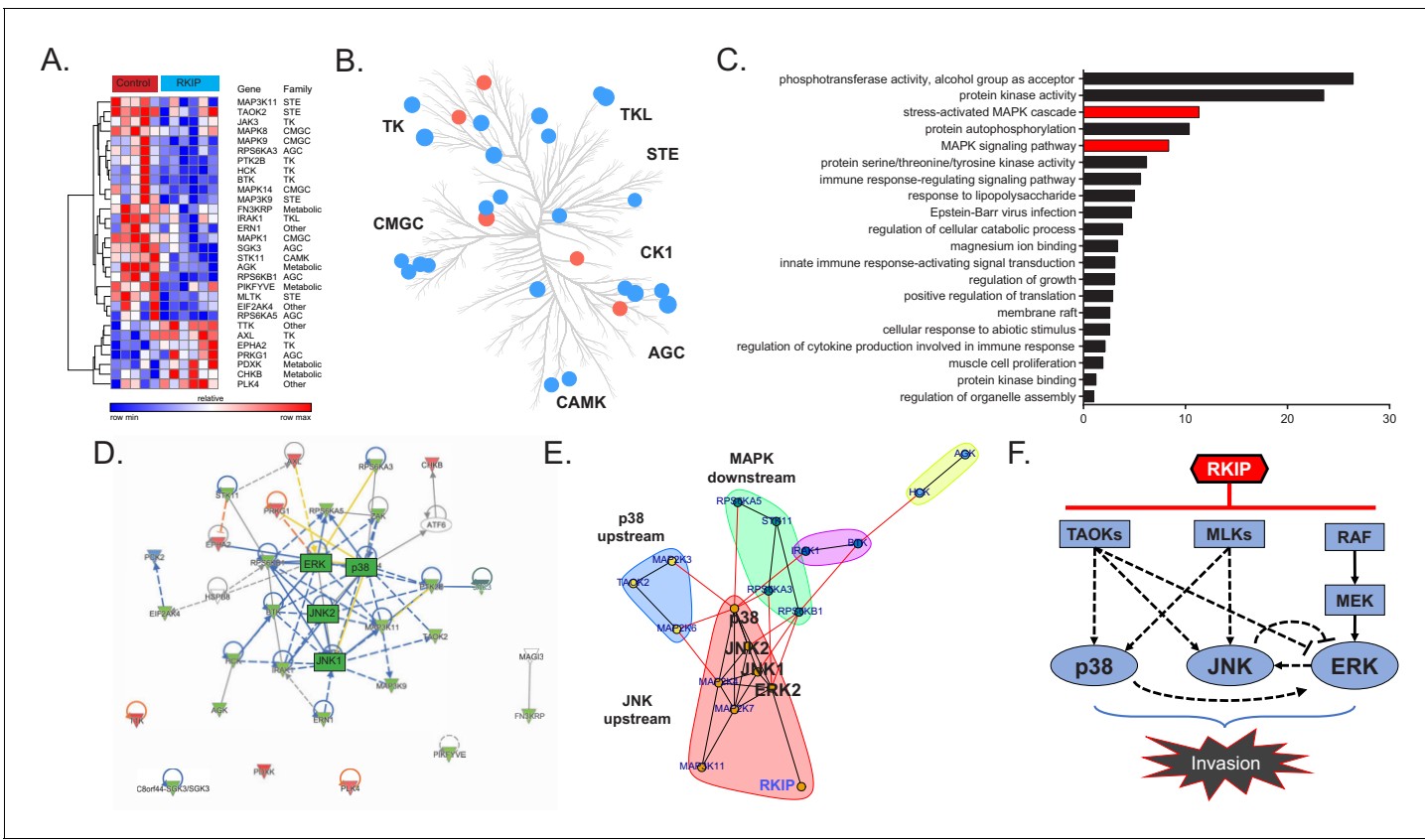

**Figure 2.** RKIP reprograms tumors by reducing signaling capacity of a network instead of targeting a single node. (A) Multiplexed inhibitor beads – mass spectrometry (MIB-MS) analysis of n = 5 control and n = 6 RKIP-expressing BM1 tumors, showing 23 kinases with reduced activity and seven kinases with enhanced activity by exogenous RKIP expression. Student's t-test, p<0.05. (B) Kinome tree displaying the distribution of kinases targeted by RKIP across different families of kinases. Blue: activity reduced by RKIP (n = 23), Red: activity enhanced by RKIP (n = 7). (C) Gene set enrichment analysis of the 23 negatively regulated kinases by Metascape. Stress-induced mitogen activated protein kinase (MAPK) related gene sets are highlighted in red. (D), Ingenuity Pathway Analysis (IPA) of the RKIP target kinases centered around MAPKs p38, JNK, and ERK. (E) Direct protein-protein interaction network and community analysis showing the core of the RKIP kinase network. (F) Diagram summarizing the interactions within the RKIP-regulated stress MAPK network in anisomycin-induced BM1 cells. Kinase interactions are determined by using small molecule inhibitors or siRNAs against the kinases in the network in three or more independent dose-response experiments with similar results (also see *Figure 2—source data 2*). The TAOK-p38 interaction is observed in cells treated with a cocktail of siRNAs against all three TAOKs (siCombo), whereas TAOK-JNK and TAOK-ERK interactions were observed by siRNAs against TAOK1 and TAOK2, respectively. For the source data, see *Figure 2—source data 1*.

The online version of this article includes the following source data for figure 2:

**Source data 1.** Source data for the MIB-MS analysis of RKIP-expressing BM1 xenograft tumors.
**Source data 2.** RKIP-regulated MAPK network displays extensive cross-talk and feedback.

Functional analysis of the downregulated kinases using Metascape (*Tripathi et al., 2015*) showed enrichment for stress kinase signaling (*Figure 2C*). Ingenuity pathway analysis indicated that most of these kinases are functionally related, and the stress MAP kinases (JNK, p38) as well as ERK comprise the core of the network (*Figure 2D*). Community analysis identified three main protein-protein interaction subnetworks within the extended MAPK family including kinases upstream of p38 (TAOK2, MKK3, MKK6), kinases upstream of JNK (MLK1, MLK3, MKK4); and p70/p90 kinases downstream of MAPKs (MSK1, RSK2, and p70/85 S6K1) (*Figure 2E*). These results indicate that the three MAPKs (JNK, p38, and ERK) as well as their upstream regulators and downstream effectors comprise the core RKIP-regulated network that drives metastasis in TNBCs.

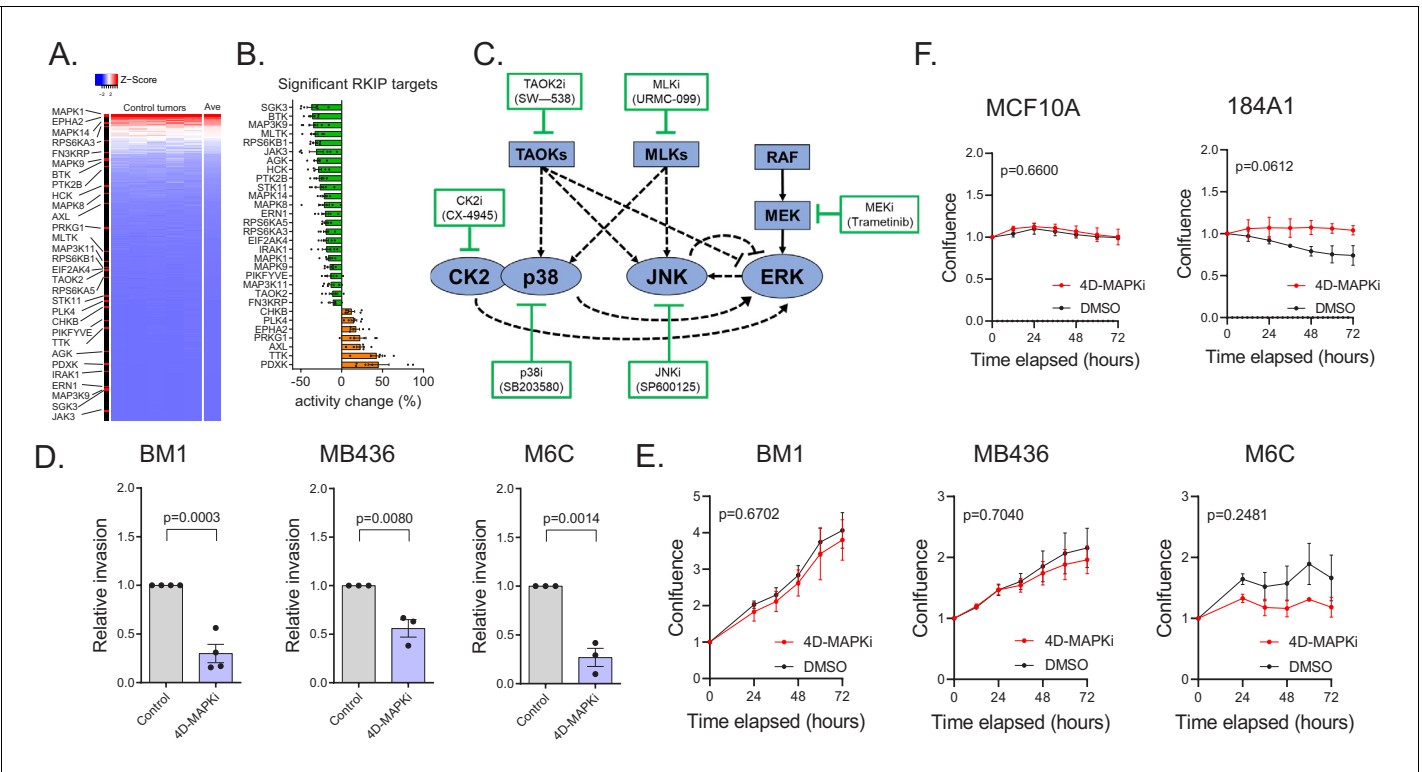

**Figure 3.** A low-dose four-drug combination reduces MAPK network signaling capacity and suppresses tumor cell invasion without altering growth. (A) Heatmap demonstrating the distribution of kinases targeted by RKIP in an activity ranked list of all 248 kinases captured. Top-to-bottom represents high-to-low ranking of MIB-captured kinases in the control cases (n = 7). (B) Percent change in kinase activity induced by RKIP in RKIP-expressing BM1 tumors with respect to the average kinase activity in the control samples. The kinases shown are the same kinases significantly regulated by RKIP according to the MIB-MS data in *Figure 2A*. (C) Diagram of the network used for modeling and the small molecule inhibitors used in the high-throughput invasion assays for potential drug combinations. (D,) Chemotactic invasion assay showing that the four-drug MAPK inhibitor combination (4D-MAPKi) blocks invasion of human and mouse TNBC cells lines. Graphs represent results from three or more independent experiments as mean ± s. e.m. Two-tailed student's t-test was used for statistical analysis. (E) 3D proliferation assay showing that 4D-MAPKi does not significantly affect the growth of TNBC cell lines. Growth curves from at least three independent experiments are represented as fold change in confluence with respect to the confluence of the cells at the time of plating, which is set to "1.0" for each experiment. For the statistical test, GraphPad Prism 9.1.0 Software's mixed effects model (equivalent of two-way repeated measures ANOVA that allows for missing values) was used. (F) 2D proliferation assays showing that 4D-MAPKi is not toxic to immortalized human mammary epithelial cell lines, MCF10A and 184A1. Data depicted as a summary of three independent experiments. Two-way ANOVA test was used for statistical analysis. For the source data, see *Figure 3—source data 1* and *Figure 3—source data 3*.

The online version of this article includes the following source data and figure supplement(s) for figure 3:

**Source data 1.** Source data for *Figure 3A,B,D,E,F*.

**Source data 2.** High-throughput invasion assays identify a low-dose four-drug MAPK inhibitor combination (4D-MAPKi).

**Source data 3.** Source data for the high-throughput IncuCyte chemotaxis invasion assays.

**Figure supplement 1.** 4D-MAPKi is more effective than the dual combinations in inhibiting all three MAPKs across multiple cell lines under anisomycin-induced stress conditions.

To understand how RKIP targets the MAPK signaling network, we identified upstream regulators of MAPKs inhibited by RKIP in tumors. MIB data from BM1 xenografts revealed three members of the MAP3K family reported to activate both p38 and JNK signaling: TAOK2, MLK1 and MLK3 (*Chen et al., 1999*; *Dhillon et al., 2007*; *Zhou et al., 2004*; *Figure 2A*). Treatment of TNBC cells with selective inhibitors or siRNAs against these MAPKKKs confirmed their ability to inhibit downstream MAPKs. Specifically, TAOKs primarily regulate p38 but may also activate JNK in TNBC cells, and MLK1,3 primarily activate JNK in TNBC cells but can also activate p38 (*Figure 2—source data 2*). MLKs and TAOKs as well as p38 and JNK have been previously shown to regulate cell motility and invasion (*Cronan et al., 2012*; *Iizuka et al., 2021*; *Koul et al., 2013*; *Sun et al., 2019*). These data suggest that RKIP inhibits the upstream TAOKs and MLKs in addition to Raf in TNBC tumors, thereby preventing activation of the pro-invasive stress MAPKs p38 and JNK in TNBC cells.

To determine the crosstalk between different MAPKs in the RKIP network, we inhibited p38, JNK, or MEK individually using small molecule inhibitors. We initially validated the efficacy of the p38 inhibitor SB203580, the JNK inhibitor SP600125, and the MEK inhibitor Trametinib using endogenous cellular targets MAPKAPK2, c-Jun, and ERK, respectively (*Figure 2—source data 2A*). Our analysis confirmed extensive crosstalk between different MAPK axes in BM1 cells (summarized in *Figure 2F*, also see *Figure 2—source data 2B*). The crosstalk involved both positive and negative feedback (e.g. the JNK-ERK axis and p38-ERK axis) suggesting that the stress kinase network regulated by RKIP is a complex network, and targeting individual MAPKs may be insufficient to mimic RKIP pharmacologically. These results indicate that, in contrast to the common therapeutic practice of fully inhibiting individual kinases at the maximum-tolerated dose, RKIP functions like a low-dose, non-toxic drug combination that reduces the activity of several key kinases within a driver network to inhibit metastasis (*Figure 2F*).

## A low-dose four-drug combination reduces MAPK network signaling capacity and suppresses tumor cell invasion without altering growth

Three aspects of RKIP regulation provide strategic guidance for anti-metastatic therapy. First, as noted above, RKIP suppresses the signaling capacity of multiple kinases within a key driver network, the stress MAPK network. Second, the kinases that are linked to metastasis ranged from high to low functional capture, indicating that the degree of kinase activity does not correlate with metastatic potential (*Figure 3A*, *Figure 3—source data 1*). Finally, the effective decrease in kinase capture for RKIP targets was generally less than 30% (*Figure 3B*, *Figure 3—source data 1*). We then determined whether we could mimic the action of RKIP using drugs to inhibit MAPK network signaling.

We sought a drug combination that, like RKIP, reduces invasion but not cell growth. We initially assessed combinations of 6 kinase inhibitors that target different nodes in the MAPK signaling network using BM1 cells (*Figure 3C*). In addition to the MEK, JNK, and p38 inhibitors used above, we also tested the MLK inhibitor URMC-099, the inhibitor CX-4945 (Silmitasertib) that blocks Casein Kinase 2 upstream of p38 and ERK signaling (*Isaeva and Mitev, 2011*; *Sayed et al., 2000*; *Zhou et al., 2016*), and the more broad-based inhibitor SW-538 that blocks kinases in the MAPK network such as TAOK2, Raf1, JNK1, HGK, and GSK3b (*Piala et al., 2016*). We monitored invasion as well as growth of cancer cells in 3D culture using a high-throughput invasion assay of nuclear-labeled BM1 cells (BM1-NucLight Red). For four drugs (p38i SB203580, JNKi SP600125, MEKi Trametinib, MLKi URMC-099), there was a minimal dose at which the proliferation rates were unaffected while the invasive capabilities of the cells were at least partially blocked (*Figure 3—source data 2*). We then tested the drugs at these minimal dosages for their combinatorial effect on invasion. Out of all the dual combinations tested, two combinations demonstrated a robust effect on invasion without a significant effect on proliferation: p38i + JNKi and MEKi +MLKi (*Figure 3—source data 2*, *Figure 3—source data 3*). Addition of a third inhibitor to these dual combinations did not improve the anti-invasive efficacy, demonstrating that the combined effect of multiple MAPK inhibitors is not necessarily additive (*Figure 3—source data 2*, *Figure 3—source data 3*).

A four-drug combination consisting of p38i, JNKi, MEKi, and MLKi (from here on referred to as 4D-MAPKi) was also successful in our initial screen at inhibiting cell invasion across multiple cell lines and stimuli without inhibiting proliferation (*Figure 3D,E*, *Figure 3—source data 2*, *Figure 3—source data 1,3*). 4D-MAPKi consistently inhibited MAPK signaling (p38,JNK,ERK) across three different TNBC cell lines, while the effect of the p38i + JNKi and MEKi +MLKi dual combinations was cell line-dependent (*Figure 3—figure supplement 1*) Notably, 4D-MAPKi was not toxic to normal

human mammary epithelial cell lines MCF10A and 184A1 (*Figure 3F*, *Figure 3—source data 1*). These findings suggest that the 4D-MAPKi combination is an effective, well-tolerated anti-invasive therapy that mimics the strategy by which RKIP inhibits the MAPK network.

## The four-drug combination suppresses metastasis, inhibits expression of pro-metastatic motility genes, and increases survival

We then determined whether 4D-MAPKi blocks metastasis of TNBC tumors in vivo. Using mouse LMB cells in a syngeneic TNBC model, we performed dose-response studies with individual drugs to determine the highest dose at which growth of the primary tumor is unaffected (*Figure 4—figure supplement 1A*). Based on this analysis, we chose a 4D-MAPKi combination of 10 mg/kg for p38i, JNKi, MLKi, and 0.5 mg/kg for MEKi (1X) for in vivo studies. At the 1X dose, the 4D-MAPKi combination significantly inhibited primary tumor growth in both syngeneic LMB tumors (*Figure 4A*) and xenograft BM1 tumors (*Figure 4B*) in a dose-dependent manner without obvious toxicity as all mice retained the same body weight (*Figure 4—figure supplement 1B*). To mitigate the confounding effect of primary tumor growth on metastasis and maximize metastatic burden, we employed tail-vein or intracardiac injection models of experimental metastasis. Both undiluted (1X) and 50% diluted (0.5X) 4D-MAPKi suppressed metastatic lung colonization in syngeneic LMB tumors in a dose-dependent manner (*Figure 4C,D*). Treatment of mice with 4D-MAPKi for only 2 days following tumor cell injection still caused significant reduction in the overall metastatic burden ~5 weeks later, suggesting that the inhibitor combination suppresses early steps of metastasis related to invasion and extravasation (*Figure 4E*). Human BM1 tumors responded better to the 4D-MAPKi treatment than LMB tumors, as even the half dose (0.5X) potently inhibited bone metastasis, the major site of metastasis following cardiac injection of these cells (*Yun et al., 2011*; *Kang et al., 2003*; *Figure 4F, G*). The 4D-MAPKi combination also improved survival of metastatic BM1-bearing mice following a 3-week treatment (*Figure 4H*). 4D-MAPKi, like RKIP, blocked induction of a significant fraction of both *Bach1* and motility genes in syngeneic LMB tumors (*Figure 4I*). These findings indicate that the low-dose drug combination, 4D-MAPKi, suppresses TNBC metastasis by reducing the signaling capacity of the stress MAPK network that transcriptionally activates metastatic genes, thereby increasing survival.

## Multi-drug combination inhibits different MAPK network topologies

To understand why the four-drug combination is so effective, we performed further experiments as well as phenomenological (observation-based) modeling of the MAPK network. Characterization of the cross-talk between different MAPKs in the RKIP-regulated network revealed that each of three human or mouse TNBC cell lines has a unique MAPK network topology (compare *Figure 2F* to *Figure 5—figure supplement 1A,B,D*). Here, we define topology as interactions between different nodes within the network that could be positive, negative, direct, or indirect. Although these network analyses are not exhaustive, our results illustrate differences in the cellular network topologies. To determine whether environmental stimulus also makes a difference in MAPK network topology, we treated BM1 cells with either anisomycin or serum, and then probed cellular response to the JNK inhibitor (JNKi). JNKi treatment did not significantly inhibit p38 in anisomycin-treated cells, whereas it actually activated p38 under serum conditions (*Figure 2F* and *Figure 5—figure supplement 1C* ). These results suggest that MAPK network topology can differ between cell lines, or even within in the same cell line based on stimulus and highlight the importance of considering network topology in determining treatment efficacy.

To understand the effect of low-dose, multi-drug treatment of metastatic cancer, we constructed a simple steady state model of the core MAPK network. Since a description based on dynamical systems theory is beyond the scope of the present work, we utilized a network flow approach (*Azeloglu and Iyengar, 2015*). Our goal was to determine the effect of introducing crosstalk into a system where a stress signal flows through three spigots (TAOK/p38, MLK/JNK and RAF/ERK pathways) and funnels into a combined output measured as BACH1 transcription. The maximal flow through the network is defined by the condition when no drugs have been administered. A node represents a kinase whose activity determines its capacity to absorb the inflow signal, and this capacity can be reduced by drug treatment. In this analysis, we assume the linear limit in which the inflow signal equals the outflow signal (*Heinrich et al., 2002*) with the kinases being in a sufficiently high

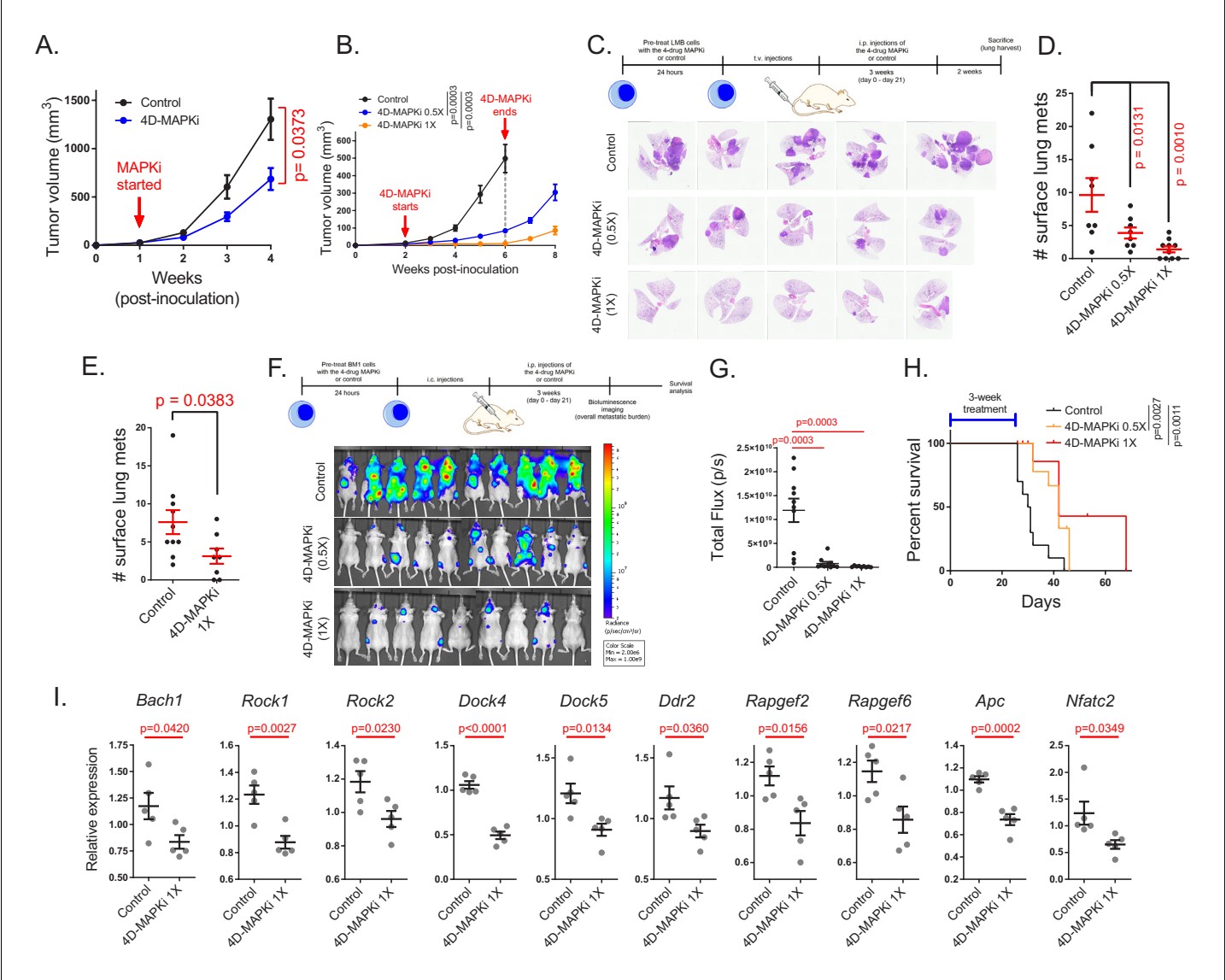

**Figure 4.** The four-drug combination suppresses metastasis, increases survival and inhibits expression of pro-metastatic motility genes. (**A**), Effect of MAPKi treatment on the primary LMB tumor growth. Mean ± s.e.m. of n = 5 biological replicates per experimental group. Two-way ANOVA test. (**B**) Primary BM1 tumor growth in mice treated with 4D-MAPKi combination for 4 weeks. Mean ± s.e.m of n = 8 control tumors, n = 8 4D-MAPKi(0.5X) treated tumors, and n = 6 4D-MAPKi(1X) treated tumors. Two-way ANOVA test at week 6. (**C**) 4D-MAPKi combination reduces metastatic tumor burden in the lungs of LMB syngeneic mouse model of TNBC. H&E staining demonstrates the metastatic lesions in cross-sections of the lungs in mice treated with 1X (undiluted, n = 10 biological replicates) 4D-MAPKi, 0.5X (diluted, n = 8) 4D-MAPKi, or the control (vehicle, n = 8). (**D**) Quantification of the visible metastatic lesions on the lung surface. Mean ± s.e.m, one-way ANOVA test with Dunnett's correction for multiple testing. (**E**) Tumor burden in the lungs of LMB syngeneic mice after 2 days (2 doses over 48 hr, on day 0 and day 1) of 4D-MAPKi treatment. Mean ± s.e.m. of n = 10 control tumors and n = 8 MAPKi(1X) treated tumors. Unpaired two-tailed student's t-test. (**F**) BM1 metastatic tumor burden in the bones of athymic nude mice treated with 4D-MAPKi at 0.5X and 1X, or control. (**G**) Quantification of the metastatic burden in (**F**). Mean ± s.e.m. of n = 10 control, n = 10 diluted 0.5 × 4 D-MAPKi, and n = 9 undiluted 1 × 4 D-MAPKi. Two-tailed student's t-test. (**H**) Overall survival of xenograft mice injected with BM1 cells via the intracardiac route after 3 weeks of 4D-MAPKi treatment. Log-rank (Mantel-Cox) test. (**I**) Expression of *Bach1* and motility genes in LMB tumors treated with 4D-MAPKi. Mean ± s.e.m. of n = 5 biological replicates per experimental group. Two-tailed student's t-test.

The online version of this article includes the following figure supplement(s) for figure 4:

**Figure supplement 1.** Effect of the individual MAPK inhibitor on tumor growth and their toxicity in combination.

concentration. A lumped representation of the signaling pathways can be built at this regime (*Beguerisse-Díaz et al., 2016*) enabling one to represent the functional networks using minimal length signaling cascades in which participants are the targetable nodes. We also assume that BACH1 expression level measurements occur when the network flow has already reached a steady state regime (*Heinrich and Rapoport, 1974*; *Kacser and Burns, 1995*). Here, we are neglecting signal amplification (*Chaves et al., 2004*) by considering that the kinase amounts are beyond the ultra-sensitivity threshold (*Huang and Ferrell, 1996*), a limit that also ensures the effectiveness of noise filtering properties (*Brandman et al., 2005*). Additionally, we assume that the cancer cells have a sufficient regular shape, and the kinases have a fast enough diffusion speed to enable us to neglect spatial effects on the signal transmission (*Friedmann, 2015*).

We utilized the anisomycin-stimulated MAPK network in BM1 cells as the basis for our model (*Figure 5A*, Network 1). If we modeled the network from other cell lines, the detailed topology would be different but the same general principles apply. We illustrate this point by generating another related MAPK network based on Network one but lacking the positive crosstalk (*Figure 5B*, Network 2). Comparison of treatment with single MAPK inhibitors (p38i, JNKi or MEKi) to 4D-MAPKi (dose restricted to ≤30% inhibition for each kinase targeted) reveals at least 60% BACH1 suppression for 4D-MAPKi and p38i in Network 1; by contrast, BACH1 levels following JNKi or MEKi treatment were only minimally reduced (*Figure 5C*). A similar result is obtained for Network 2, although in this case JNKi has no significant effect on BACH1 levels (*Figure 5D*). These results from our models suggest that, while response to individual inhibitors may vary because of differences in network topology, 4D-MAPKi is more robust in inhibiting network output across different topologies.

To test this prediction, we assessed *BACH1* gene expression as a measure of network output in the three TNBC cell types with different network topologies. Dose-response studies with individual MAPK inhibitors showed that *BACH1* expression can be regulated by either ERK, JNK or p38 in at least one TNBC cell line (*Figure 5E–G*, *Figure 5—source data 1*). However, as our model predicted, no single MAPK inhibitor even at maximum dose effectively reduced *BACH1* expression across the three cell lines (*Figure 5H–J*, left panels and *Figure 5—source data 1*). By contrast, the low-dose 4D-MAPKi combination attenuated *BACH1* gene expression in all human and mouse cell lines tested (*Figure 5H–J* right panels, *Figure 5—figure supplement 2A*, *Figure 5—source data 1*). Consistent with the in vitro data for the mouse LMB cell line (*Figure 5G,J*), both 4D-MAPKi and MEKi treatment significantly reduced metastatic burden in the lungs of mice with orthotopic LMB tumors, while p38i showed no effect (*Figure 5—figure supplement 2B*). These data indicate that 4D-MAPKi more effectively regulates *BACH1* expression across cells with different MAPK network topologies than single high-dose inhibitors.

## Limiting the extent of kinase inhibition at multiple nodes reduces network output and prevents compensatory network activation

Maximum tolerated dose regimens can yield promising responses in certain patients. (*Gallaher et al., 2018*; *Duncan et al., 2012*; *Wong et al., 2019*) In terms of our mathematical model, when all pathways are independent and operating at their maximal capacity, reducing or eliminating any one of them restricts the output flow. In this scenario, a single target therapy can potentially be effective at reducing output. However, if flow from the single node is eliminated, then excess flow from the initial functional network could end up activating a different functional network, leading to a compensatory increase in overall output (*Seton-Rogers, 2014*) (see CKN, compensatory kinase network in *Figure 6A*). Even a small reduction in the activity of multiple nodes of a given pathway, although insufficient for eliminating the activity of the functional network, would reduce the overall surplus, and hence, the chances for compensatory activation.

To address the role of surplus signal flow following inhibitor treatment, we determined whether different degrees of inhibition of MEK would yield different outcomes with respect to activation of compensatory driver networks. As a model system, we analyzed induction of an epidermal growth factor (EGF) receptor-PI3K feedback loop in response to MEK inhibition of EGF-stimulated BM1 cells. Activation of this compensatory PI3K pathway was previously reported to restrict the efficacy of MEK inhibitors in basal subtypes of breast cancer (*Mirzoeva et al., 2009*). Following maximal inhibition of MEK in our model, the surplus flow from Raf approaches 30% of the initial input signal that can be funneled to a compensatory kinase network such as PI3K/AKT (*Figure 6B*). When we modeled 4D-MAPKi, on the other hand, it only diverted ~10% of the incoming stress signal to a

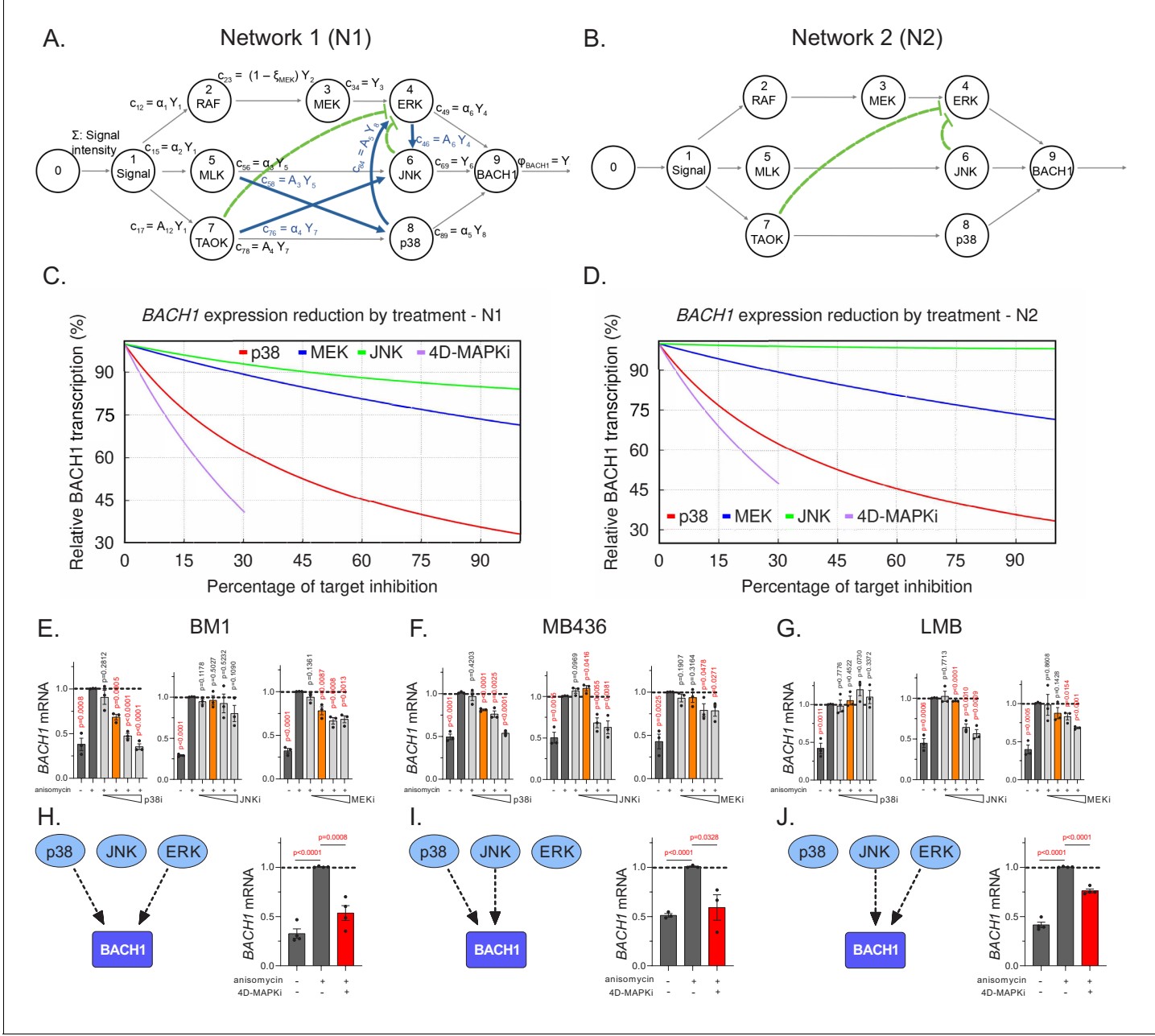

**Figure 5.** Multi-drug combination inhibits different MAPK network topologies. (**A**), The topology of the stress response of the core MAPK driver network activating *BACH1* gene transcription, as output, in BM1 cells. This network topology, termed N1, is composed of multiple kinase signaling pathways responsible for activating *BACH1*. The nodes of the signaling network, represented by circles, are kinases within the network. The arrows directed toward nodes indicate the inflow from a signal or an active kinase at the upstream node. The product of a node, resulting from the interaction between the upstream signal and downstream kinases, is denoted by arrows leaving the node. The pathways are indicated by black arrows while the crosstalk between different pathways is denoted by blue and green arrows. The non-linear repression of one node by another is represented by the green lines with bars directed toward the repressed component. (**B**) A hypothetical BACH1 stress response driver network, denoted as N2, that has no crosstalk between its individual pathways. The interpretation of its symbols is the same as in **A**. (**C**) Graph depicting predicted downregulation of *BACH1* following cell treatment with specific inhibitors relative to maximal stress-induced *BACH1* gene transcription in cells with N1. The y axis shows the percentage of maximal *BACH1* gene transcription, and the x axis denotes the percent inhibition of each kinase targeted by a drug or drug combo relative to the maximal inhibitor dose (set at 100% inhibition). Relative *BACH1* gene transcription in response to p38i, MEKi, JNKi, or the 4D-MAPKi drug combo is indicated in red, blue, green, or purple. (**D**) Graph depicting predicted downregulation of *BACH1* following cell treatment with specific inhibitors relative to maximal stress-induced *BACH1* gene transcription in cells with N2. The y axis shows the percentage of maximal *BACH1* gene transcription, and the x axis denotes the percent inhibition of each kinase targeted by a drug or drug combo relative to the maximal inhibitor dose (set at 100% inhibition). Relative *BACH1* transcription in response to p38i, MEKi, JNKi, or the 4D-MAPKi drug combo is indicated in red, blue, green, or

*Figure 5 continued on next page*

*Figure 5 continued*

purple. (**E–G**) Single agent dose-response experiments demonstrating that *BACH1* expression is activated by different MAPKs in different cell lines. Orange bars indicate the final dosage of an individual inhibitor used in 4D-MAPKi. (**H–J**) The network-targeting 4D-MAPKi is able to decrease *BACH1* expression across all three cell lines even though *BACH1* is regulated by a different set of MAPKs in each cell line. Left panels: Diagrams summarizing *BACH1* regulation by MAPKs in each TNBC cell line. Right panels: 4D-MAPKi blocks *BACH1* mRNA expression in anisomycin-induced cells. For **E-J**, the bar-graphs represent three or more independent experiments performed in each cell line, where the *BACH1* expression in drug-treated cells is measured with respect to aniso-induced non-treated positive control group. Statistical significance for each dose was determined by student's t-test with respect to the positive control group. For the source data, see *Figure 5—source data 1*.

The online version of this article includes the following source data and figure supplement(s) for figure 5:

**Source data 1.** Source data for *Figure 5E–J* and *Figure 5—figure supplement 2A* .
**Figure supplement 1.** Differences in MAPK network topology among cell lines.
**Figure supplement 2.** The effect of 4D-MAPKi on MAPK network output.

compensatory kinase network (CKN) because we restricted MEK/ERK inhibition to less than 30%. This type of analysis, carried out for each of the kinases in the MAPK network, shows that the combined surplus of kinase signal following 4D-MAPKi inhibition approaches 60% (*Figure 6C*). However, this signal is dissipated among multiple kinases such that the resulting surplus from each kinase is insufficient to activate compensatory networks.

To test these predictions experimentally, we generated dose response curves to assess the relationship between degree of MEK inhibition (assessed by p-ERK) and induction of PI3K (assessed by p-AKT). The results show that, under controlled in vitro conditions, AKT activation has a threshold effect with minimal change in activity until ERK is inhibited by ~30% (*Figure 6D*, *Figure 6—source data 1*). Consistent with this finding, when we tested 4D-MAPKi in EGF-stimulated cells, ERK signaling was robustly inhibited (~75%) and AKT was maximally activated (*Figure 6—figure supplement 1A*; *Figure 6—figure supplement 1B*). By contrast, in serum-stimulated cells, 4D-MAPKi inhibited ERK by only ~40% and AKT activity was actually reduced (*Figure 6—figure supplement 1A*; *Figure 6—figure supplement 1B*, *Figure 6—source data 1*). We also checked the activation of p-AKT (Ser473) in the syngeneic LMB primary tumors treated with either the vehicle control or the 4D-MAPKi regimen. We observed no significant activation of p-AKT signaling in the primary tumors treated with 4D-MAPKi (*Figure 6—figure supplement 1C*), indicating that the compensatory pathway activation is not triggered. Taken together, these results show that targeting multiple kinases in a network or limiting kinase inhibition, as RKIP does, is an effective mechanism to avoid compensatory AKT activation above background.

These experimental and mathematical analyses suggest that effective inhibition can be accomplished by targeting several nodes belonging to different pathways within the same driver network. This will reduce the flow through multiple pathways of the network and its resulting output, decreasing the efficacy of the driver network. Reduction of overall output, however, creates a surplus signal that cannot be accommodated by other kinases within the network and, instead, is directed toward compensatory driver networks. Thus, it is important to keep the overall surplus dissipated among multiple nodes, and the surplus signal from each kinase below the threshold for activation of its compensatory network. Of note, since our goal is to suppress metastasis, this partial inhibition which leaves the growth network largely intact is still effective. Together, these studies suggest that (1) multi-kinase targeting is more effective than single kinase inhibition across different cells and environmental stimuli; and (2) low inhibitor doses are less likely than high inhibitor doses to trigger feedback activation of compensatory networks.

## The BACH1/motility gene axis, targeted by RKIP and 4D-MAPKi, is associated with multiple cancers and metastasis suppressors

To understand the clinical significance of the MAPK network suppressor (RKIP) and the MAPK network output (*BACH1* and motility-related genes), we looked at their relative gene expression in the TCGA database. Remarkably, stratifying breast cancer patients by high RKIP (*PEBP1*) and low *BACH1* expression or vice versa reveals a striking inverse association of RKIP with *BACH1* and the motility-related genes in ~60% of patients (*Figure 7A*). These data suggest that the RKIP/*BACH1*/motility gene axis identifies breast cancer patients who would be therapeutic candidates for 4D-MAPKi treatment. Enrichment of the motility-related target genes identified in the present study

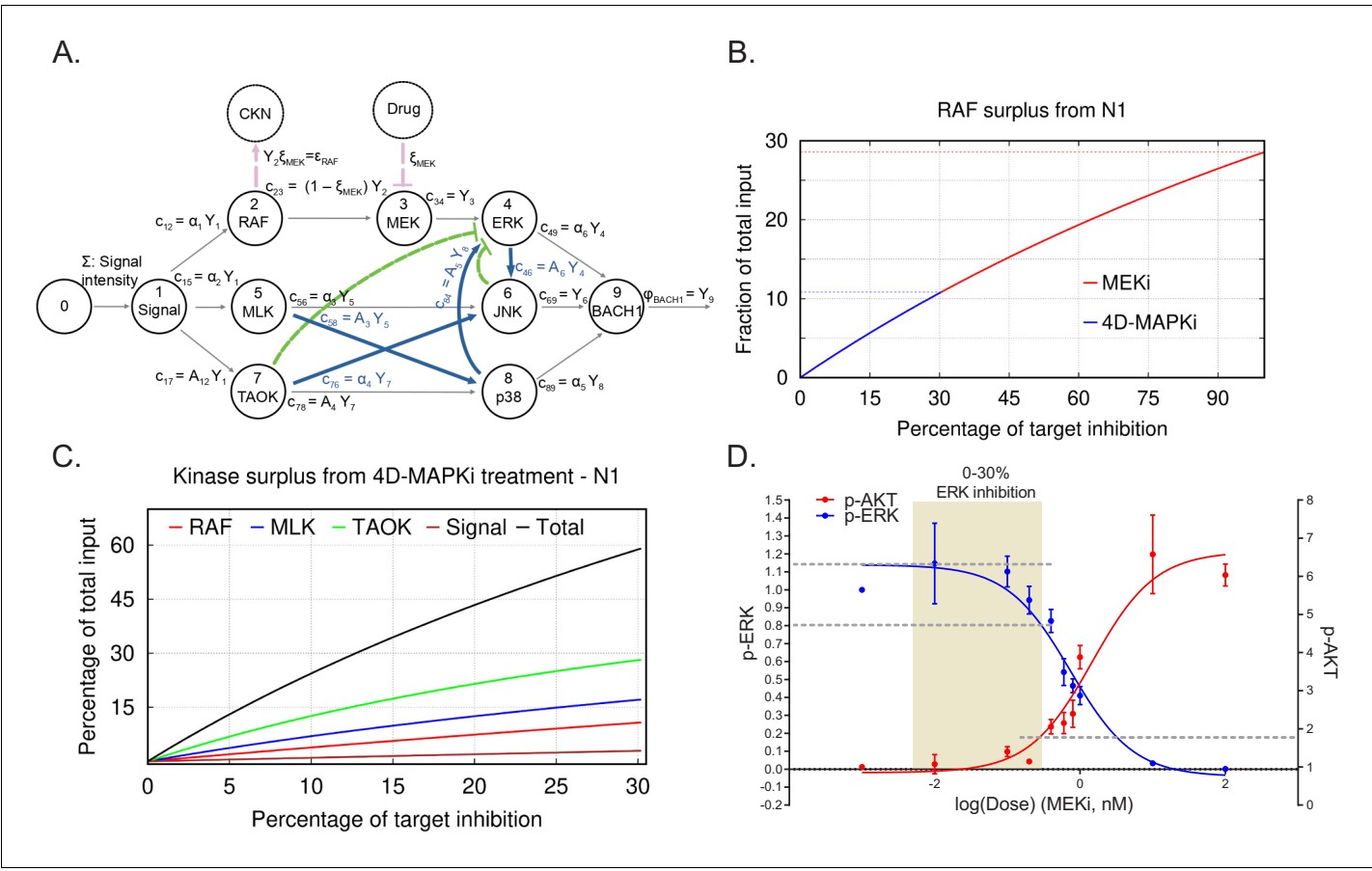

**Figure 6.** Limiting kinase inhibition at multiple nodes reduces network output and prevents compensatory network activation. (**A**) N1 illustrating the case of a treatment (MEKi) targeting node 3 (MEK) and activation of a compensatory kinase network (CKN) linked to upstream node 2 (RAF) (red circle). A comparable diagram can be generated for each upstream node to describe surplus signal activating a distinctive CKN. The percentage of reduction on the activity at a target node because of a treatment dose $x$ is indicated by $\xi$. This inhibition causes reduction in the flow of product to node 3, denoted by $c_{23}=(1-\xi_{MEK})Y_2$ (see **A**). (**B**) Graph depicting surplus signal from RAF as a fraction of the total input signal $\Sigma$ following treatment by MEKi or 4D-MAPKi of cells with N1. The x axis denotes the percent inhibition of each kinase targeted by inhibitors relative to the maximal inhibitor dose (set at 100% inhibition). The y axis is the fraction of surplus RAF signal generated following drug treatment of cells relative to total input signal. MEKi, red; 4D-MAPKi, blue. (**C**) Graph depicting surplus kinase signal as a fraction of the total input $\Sigma$ following treatment by 4D-MAPKi of cells with N1. The surplus is a consequence of the congestion of each direct pathway causing an insufficient absorption of the stress input by the driver network and its redirection toward a compensatory network. The x axis denotes the percent inhibition of each kinase targeted by 4D-MAPKi relative to the maximal inhibitor dose (set at 100% inhibition). The y axis is the fraction of surplus kinase signal generated following 4D-MAPKi treatment of cells relative to total input signal. RAF (surplus from MEKi), red; MLK (surplus from JNKi), blue; TAOK (surplus from p38i), green; and Signal (surplus from MLKi), brown; Total (sum of all surplus signals), black. (**D**) Dose-response curves showing activation of the compensatory PI3K network, monitored by p-AKT levels, when EGF-induced BM1 cells are treated with increasing doses of MEKi. Mean ± s.e.m. of n = 3 independent experiments. For the source data, see *Figure 6—source data 1*.

The online version of this article includes the following source data and figure supplement(s) for figure 6:

**Source data 1.** Source data for *Figure 6D* and ; *Figure 6—figure supplement 1B*.

**Figure supplement 1.** Effect of 4D-MAPKi on MAPK compensatory network activation.

extended beyond breast cancer. The same gene families were also inversely correlated with RKIP in other solid TCGA cancer types (*Figure 7B,C*) including pancreatic, ovarian, lung, head and neck, and colorectal. Of note, in cancers where no correlation to these specific motility-related genes was observed, we noticed a strong correlation to other members of the same gene families (*Figure 7C*). Finally, expression of other experimentally validated metastasis suppressors (*BRMS1*, *ARGHDIA*, *NME1*, and *DRG1*) (*Zhao et al., 2015*) also negatively correlated with motility-related gene sets (*Figure 7D,E*). These clinical analyses suggest that BACH1-regulated motility-related machinery is a hallmark of metastasis that is targeted by multiple physiological suppressors such as RKIP.

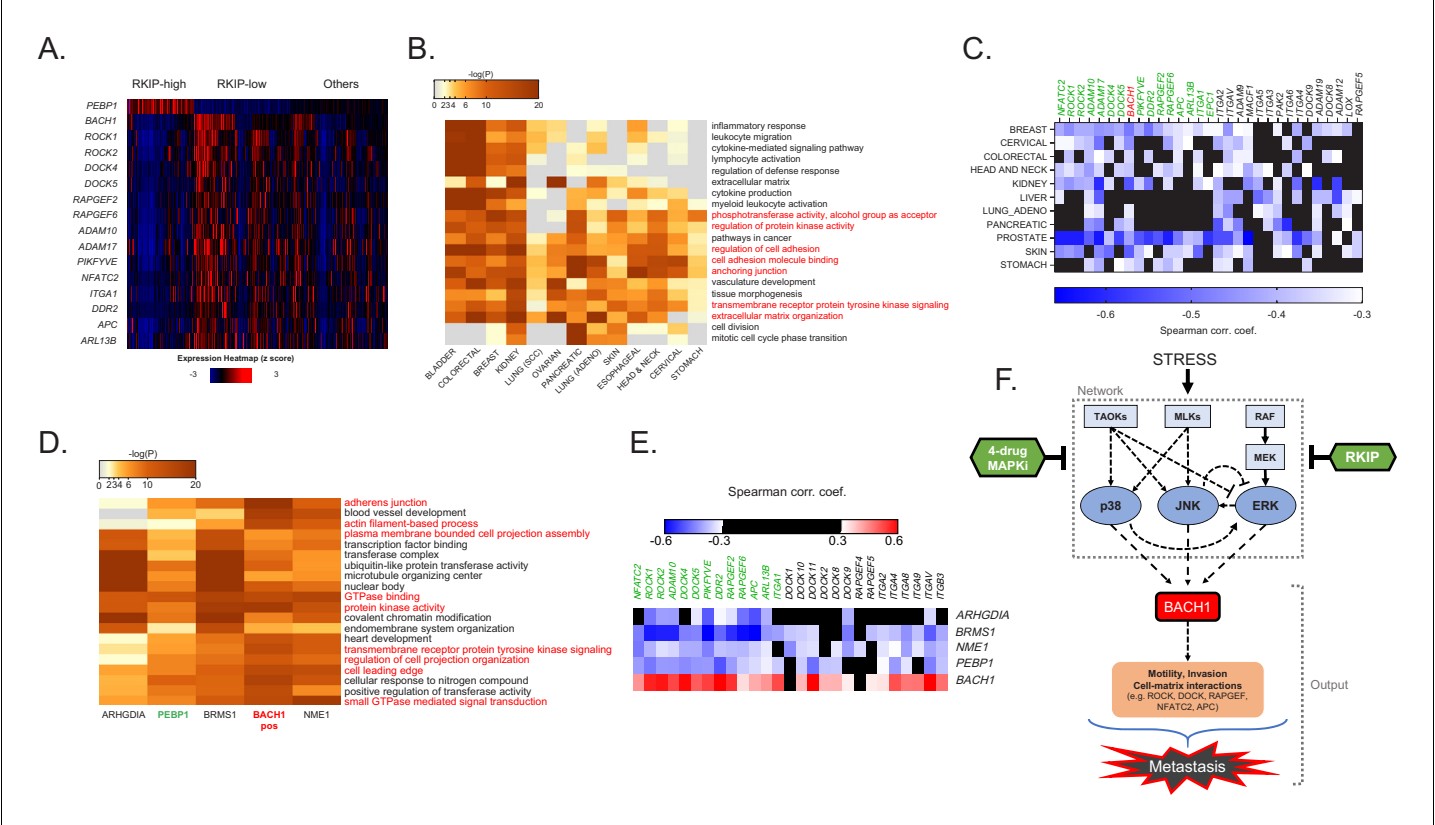

**Figure 7.** The *BACH1*/motility gene axis, targeted by RKIP and 4D-MAPKi, is associated with multiple cancers and metastasis suppressors. (**A**) Expression of RKIP (*PEBP1*), *BACH1*, and the downstream motility genes in each TCGA BRCA patient (n = 1100), grouped by RKIP status. RKIP-high: z-score > 0.5 (n = 274), RKIP-low: z-score < −0.5 (n = 414), Others: −0.5 < z score<0.5 (n = 412). (**B**) Gene sets enriched with genes negatively correlated with RKIP across multiple TCGA cancer types. (**C**) Spearman correlation coefficients for *BACH1* or motility genes relative to RKIP in TCGA cancers. Coefficient cutoff of −0.3 (coefficients between −0.3 and 0 were colored black). (**D**) Motility-related gene sets enriched for genes that negatively correlate with the indicated metastasis suppressors, but positively correlate with *BACH1* in the TCGA BRCA set. (**E**) Spearman correlation coefficients for *BACH1* or motility genes relative to RKIP (*PEBP1*) or other metastasis suppressors in TCGA BRCA. Coefficient cutoff of 0.3 in both positive and negative side. Coefficients between −0.3 and 0.3 were colored black. (**F**) Diagram summarizing the stress MAPK kinase N1 network that regulates metastasis in breast cancer. Stress activates a network of MAPKs that interact via crosstalk. RKIP and the RKIP-mimicking drug combo 4D-MAPKi reduce the signaling capacity of the entire network by targeting multiple nodes. This allows for effective reduction of the metastatic output of the network, measured by the expression of pro-metastatic *BACH1* and its target motility genes.

Taken together, our results show that stress as an input activates the core MAPK network, leading to induction of *BACH1* expression as an output which in turn activates motility-related genes required for invasion and metastasis (*Figure 7F*). Inhibition of this network by metastasis suppressors such as RKIP or a low-dose four-drug combination effectively restricts expression of invasive genes and reduces the metastatic phenotype.

## Discussion

Here, we propose a new approach for therapeutic targeting of metastatic disease based upon the action of physiological metastasis suppressors such as RKIP. In this study, we focus on the underlying general principles of suppressor action that enable stable metastatic inhibition without triggering activation of compensatory driver networks. Instead of completely inhibiting specific nodes, suppressors reduce the signaling capacity of a driver network by partially targeting multiple kinases within the network, thereby restricting the output that promotes invasion and metastasis. We experimentally validate this approach using a four-drug combination that acts on the core MAPK network to inhibit metastasis and promote survival in mouse TNBC models. This approach even works for

different driver network topologies associated with diverse cells within tumors, conferring a higher degree of robustness to our therapeutic strategy.

Mathematical modeling using a simple steady state model illustrates the underlying concept. The model suggests that multidrug combinations are more effective at suppressing signaling across cells with different MAPK network topologies. In addition, minimizing the inhibition of each kinase enables dissipation of excess signal so that compensatory networks are not activated. Indeed, our experimental findings support these general principles, demonstrating that different TNBC cells have different MAPK network topologies and respond in diverse ways to MAPK inhibitors. Our data demonstrate that recurrence through activation of p-AKT after treatment of cells with MEK inhibitor is a threshold effect, and p-AKT is not induced by the four-drug combination in mice, further supporting the relevance of our proposal and opening a new avenue of investigation aimed at finding the range of topologies within a single driver network and their relationship to multiple compensatory driver networks.

Despite the relevance of inferring the uncertainty of a topology of a functional network, such a quantification is beyond the scope of the current manuscript. That would demand a focused investigation based on Bayesian inference (*Gelman et al., 2013*). This is a non-trivial task, however, as even Bayesian inference of the kinetic constants of the law of mass action used on determination of the minimal set of participants of a given signaling pathway is an active research field (*Vanlier et al., 2013*; *Liepe et al., 2010*; *Liepe et al., 2014*; *Vyshemirsky and Girolami, 2008*; *Xu, 2010*). Here, we aim to develop a framework for inferring the topology of functional networks composed of multiple pathways. The experimental determination of kinetic constants of multiple pathways or their estimation based on Monte Carlo simulations are challenging tasks (*Miskovic et al., 2019*) that will require the development of more efficient computational and theoretical techniques.

Effects on primary tumor growth can be confounding when monitoring spontaneous metastasis. Therefore, we utilized two different models to illustrate the effect of the four-drug combination on metastasis. When using the spontaneous metastasis model where metastasis is derived from the primary tumor, it is difficult to distinguish inhibitory effects on metastasis alone from inhibitory effects on primary tumor growth. All cells that grow at either primary or metastatic sites have a survival component, so we are assuming here that survival at the metastatic site is an aspect of metastasis. We successfully distinguished growth from invasion in vitro and, using a similar strategy, identified drug concentrations that individually did not inhibit growth in vivo. However, since drug effects are usually not additive, it is challenging to precisely titrate the four drugs so that we suppress metastasis but do not affect primary tumor growth in vivo. Therefore, as an alternative assay that we and others have used in the past to monitor metastasis alone (e.g. *Yun et al., 2011*), we utilized the tail-vein or cardiac injection model that focuses on extravasation and metastatic colonization using comparable numbers of tumor cells. These assays clearly show a robust inhibition by the four-drug combination when the same number of cells are present in the circulation. Even a limited 2-day treatment with drug after tail vein injection suppressed metastatic growth, suggesting that 4D-MAPKi primarily affects the early steps of metastatic seeding such as extravasation, invasion, or colonization. Ultimately, our goal is to suppress metastasis as well as growth of the primary tumor. The 4D-MAPKi we identified can act as a rheostat, mimicking a metastasis suppressor at low dose but also inhibiting growth if higher doses are used.

Using databases such as TCGA combined with preclinical studies enables identification of metastasis suppressor-regulated genes that can be used as biomarkers both for development of drug combinations and identification of patients who would benefit from those therapies. Similar correlations between expression of other metastasis suppressors and motility-related genes in other tumor types suggest that this approach can have wide application to pharmaceutical treatment of metastatic disease states in multiple tissues. In particular, our findings suggest that 4D-MAPKi would be an effective treatment for metastatic breast tumors that express the motility-related genes but lack suppressors that inhibit them. We anticipate that, in addition to identifying 4D-MAPKi, this approach can be utilized to discover other multi-drug combinations that are effective.

In at least one tumor model, our data shows that the MEK inhibitor is as efficacious at suppressing metastasis in vivo as the four-drug combination. Our studies suggest, however, that the four-drug regimen is more likely to have efficacy across different cellular networks and different environmental conditions and therefore work better when treating heterogeneous tumors. We explored this concept by comparing in vivo efficacy for single agent MEK or the other inhibitors (p38, JNK, MLK)

to the four-drug regimen using the syngeneic LMB mouse model. When we tested the drugs on LMB cells in culture, the MEK and JNK inhibitors and the four-drug combination were effective at suppressing MAPK network output (assessed by the pro-metastatic gene BACH1) whereas the other inhibitors were not. Like the cell culture studies, both MEK and four-drug regimens reduced metastatic burden in the lungs of the mice but p38 inhibitors did not. The fact that JNK inhibitors inhibited *BACH1* gene expression in vitro but did not suppress metastasis in vivo could be due to a number of factors including drug accessibility, different outputs, or microenvironmental interactions. Taken together, these results support our rationale for utilizing multi-drug combinations. In addition, it should be noted that, while MEK inhibitors as single agents can be very effective in blocking tumor growth and metastasis in preclinical settings, they rarely work long term in clinical settings because of the compensatory activation of resistance pathways such as PI3K/AKT signaling. Drugs such as these, when used as single agents, have not given durable responses thereby necessitating multi-drug approaches (*Menzies and Long, 2014*). Our studies suggest that 4D-MAPKi may exhibit similar or better efficacy to MEK inhibitors without activating these compensatory pathways.

4D-MAPKi, which targets MEK, p38, JNK, and MLK, is a novel and clinically feasible multi-drug combination (*Messoussi et al., 2014*; *Okada et al., 2017*; *Mora Vidal et al., 2018*; *Cicenas et al., 2017*). Data already available from MEK inhibitor Phase 1 trials (*Infante et al., 2012*) should enable an estimate of drug doses that would reduce MEK activity by less than 30%, and similar analyses can be carried out for the other kinase inhibitors. Effective treatment based on inducing functional RKIP in tumors has not been possible since RKIP regulation is complex and occurs at multiple levels including transcriptional, translational, and post-translational (*Yesilkanal and Rosner, 2018*). By significantly expanding our knowledge of RKIP function as a tumor metastasis suppressor, this study identifies additional therapeutic targets. MLKs and TAOKs, as regulators of p38 and JNK, have not previously been implicated as a part of the RKIP-network. BACH1, which we recently identified as an inhibitor of mitochondrial metabolism that can be targeted independently in ~30% of breast tumors (*Lee et al., 2019*), is shown here to be a transcriptionally-regulated and pro-invasive mediator of the stress MAPK network.

While the utility of multi-drug combinations for therapeutic treatment is widely acknowledged, the strategy proposed here differs significantly in several respects. First, we focus on reducing signaling capacity in one driver network rather than fully inhibiting single nodes belonging to one pathway. Second, our findings argue that targeting multiple nodes associated with distinct pathways within the same network is more likely to improve response and prevent compensatory signaling across a heterogeneous tumor cell population than targeting several single nodes distributed among different cellular networks (reviewed in *Smith and Wellbrock, 2016*). Third, by restricting the extent of inhibition at each single node, we dissipate excess signal flow to avoid activation of compensatory networks that promote resistance and recurrence. The common strategy of maximal inhibition at multiple nodes is more likely to lead to resistance since the surplus may cross the activation threshold of a compensatory driver network. Finally, our goal here is to suppress metastasis, the process responsible for cancer lethality, rather than primary tumor growth.

During revision of this manuscript, low-dose kinase inhibitor combinations targeting the EGFR-RAS-RAF-MEK-ERK pathway have been shown in pancreatic cancer and NSCLC to be effective at inhibiting primary tumor growth and inducing apoptosis (*Fernandes Neto et al., 2020*; *Ozkan-Dagliyan et al., 2020*), similar to our findings that targeting p38-JNK-ERK MAPKs with low-dose treatments will suppress metastasis and, in some cases, tumor growth. While these additional studies support our argument for using low-dose multi-drug combinations, our approach of targeting a functional driver network differs from the vertical inhibition of the linear EGFR-RAS-RAF-MEK-ERK pathway, as it is less vulnerable to mutational resistance within a single pathway as well as compensatory signaling mechanisms. Although small decreases in the activity of multiple nodes within a given pathway would not eliminate the activity of the functional network, they should reduce the overall surplus and the likelihood of compensatory activation.

By first inhibiting metastasis and associated cellular heterogeneity, we anticipate that subsequent treatment with even traditional cytotoxic agents as radio-, chemo-, or immunotherapy will be more effective. Metastatic progression is a dynamic and highly drug-resistant process. While early metastatic seeding can take place before the primary tumor is clinically detectable, the primary tumor continuously sheds metastatic cells into the circulation that can form metastases at other sites (*Quinn et al., 2021*; *Kim et al., 2009*; *Gupta and Massagué, 2006*). Therefore, an anti-metastatic

treatment that slows down this dynamic spread would have therapeutic benefit even at a late stage of disease. In addition, evidence also suggests that metastatic cells are more resistant to systemic treatments due to their more mesenchymal phenotype, and low proliferation rates (*Boston Change Process Study Group, 2005*). If an anti-metastatic therapy such as 4D-MAPKi can revert metastatic cancer cells back to a less metastatic, more epithelial-like state, it can sensitize these cells to certain systemic and metabolic treatments as we have previously shown (*Lee et al., 2019*). Therefore, we suggest a two-part strategy to convert metastatic cells to a non-metastatic state prior to treatment with agents that will kill proliferating cells. As such, this strategy represents a paradigm shift in how we address treatment of metastatic disease in cancer.

## Materials and methods

### Cell lines

In our studies, we used the human TNBC cell lines BM1 (also known as MDA-MB-231-BM1, BoM1, 1833) and MB436 (also known as MDA-MB-436), the mouse TNBC cell lines LMB (also known as E0771-LMB) and M6C, the immortalized normal mammary epithelial cell lines MCF10A and 184A1, and the human embryonic kidney epithelial cell line 293T. MB436, MCF10A, 184A1, and 293 T cells were received from American Type Culture Collection (ATCC). E0771-LMB (LMB) cells were generated by Robin Anderson (*Johnstone et al., 2015*). M6C cells were generated by Jeffrey Green (*Holzer et al., 2003*). BM1 cells were generated by Massagué and colleagues (*Kang et al., 2003*). BM1, MB436, LMB, and M6C cells were cultured in DMEM media with 10% fetal bovine serum (FBS), penicillin (50 U/ml), and streptomycin (50 µg/ml). MCF10A and 184A1 cells were grown in DMEM/F-12 (50/50) with 10% FBS and penicillin-streptomycin. All cell lines were authenticated by short tandem repeat analysis and used within 15 passages after their arrival in the laboratory. Mycoplasma detection was routinely performed to ensure cells were not infected with mycoplasma using MycoAlert Detection kit (Lonza, LT07-218).

### Small molecule inhibitors

For in vitro and in vivo studies, JNK inhibitor SP600125, MEK inhibitor Trametinib (GSK1120212), MLK inhibitor URMC-099, and CK2 inhibitor CX-4945 (Silmitasertib) were purchased from APExBIO (A4604, A3018, B4877, A833010, respectively). p38 inhibitor SB203580 was purchased from Selleckchem (Cat No S1076) for the in vitro experiments. For in vivo studies, water soluble SB203580 hydrochloride was purchased from APExBIO (B1285). SW-538 (SW034538) was provided by Elizabeth J. Goldsmith et al.

### Signaling studies in vitro

In order to study the changes in stress kinase signaling upon stress in the presence of RKIP, or small molecule inhibitors of the MAPKs, cells were plated at sub-confluence. Once they reach roughly 70% confluence, they were starved overnight (16–24 hr) in serum-free media, and then induced with anisomycin (Sigma-Aldrich, Cat No A9789) at 25 ng/ml final concentration or 10% serum for 30 min to activate MAPK pathways. In studies with small molecule inhibitors of the MAPK pathway, all inhibitors were re-suspended in DMSO and used at indicated concentrations. The cells were pre-treated with the inhibitors in serum-free media for 30 min after overnight serum starvation, immediately before induction with anisomycin or serum for 30 min. In this case, the inducing agent was directly added to the pre-treatment media that already had the inhibitors, or the pre-treatment mediate was replaced by fresh media containing the inducer and the inhibitors. This is to ensure the inhibitors are present during induction of the MAPK pathways. Upon induction for 30 min, the cells were washed three time with cold PBS and immediately lysed in RIPA buffer for protein collection.

### Protein isolation and western blots

Cultured cells were washed with cold PBS and lysed in RIPA buffer with protease inhibitors (Millipore Sigma, 539134) and phosphatase inhibitors (GoldBio, GB-450). Tumor samples were snap-frozen in liquid nitrogen, pulverized, and lysed in RIPA buffer with protease and phosphatase inhibitors. All samples were sonicated three times for 10 s at 35% power and centrifuged at max speed for 15 min at 4°C. Supernatant was collected and the protein concentration was measured using the Bradford

assay. All samples were boiled in 6X Laemmli buffer immediately after protein concentration measurement.

For western blots, equal amounts of protein, ranging from 10 μg to 50 μg, across all samples were used. Blots were blocked for 1 hr at ambient temperature with either Odyssey Blocking Buffer (LI-COR Biosciences, 927–40010, diluted 1:1 with PBS) or with 5% FBS in Tris Buffer Saline (TBS) with 0.1% Tween20. Then, blots were incubated with primary antibodies at 4˚C over-night, and with secondary antibodies at ambient temperature for 1 hr. Finally, blots were treated with ECL reagent (Pierce ECL Western Blotting Substrate, Thermo Scientific, 32106) when HRP-conjugated secondary antibodies were used and developed under the Chemiluminescence channel of the LI-COR Fc Imaging System. Blots with fluorescent secondary antibodies were imaged under 700 nm or 800 nm channels of LI-COR Fc. Signal intensity was quantified using Image Studio Lite (LI-COR) software.

## Primary antibodies used

Phospho-TAOK3 (Ser177) +Phospho TAOK2 (Ser181) +Phospho-TAOK1 (Ser181) (Abcam, ab124841)
Phopsho-p44/42 MAPK (ERK1/2)(Thr202/Tyr204) (Cell Signaling, 9101)
Phospho-SAPK/JNK (Thr183/Tyr185) (Cell Signaling, 9251)
Phospho-p38 MAPK (Thr180/Tyr182) (Cell Signaling, 4511)
Phospho-AKT1 (S473) (Cell Signaling, 4060) alpha-Tubulin (Santa Cruz, sc-8035) alpha-Tubulin (Invitrogen, MA1-19401)
GAPDH (Santa Cruz, sc-32233)
BACH1 (Santa Cruz, sc-271211)

## Secondary antibodies used

Goat anti-Mouse IgG (LI-COR, IRDye 800CW, 926–32210)
Goat anti-Mouse IgM (LI-COR, IRDye 800CW, 926–32280)
Goat anti-Rabbit IgG (LI-COR, IRDye 680RD, 926–68071)
Goat anti-Rabbit IgG, HRP conjugate (EMD Millipore, AP187P)
Goat anti-Mouse IgM, HRP conjugate (Invitrogen, 31440)
Goat anti-Mouse IgG, HRP conjugate (Sigma Aldrich, A4416)

## Transient transfection

Prior to transfection, the cells were plated in six-well plates and grown to ~70% confluence. siRNA vectors were used at a final concentration of 50 nM per well of cells. The vectors were incubated with 10 μl of Lipofectamine 3000 (Invitrogen, L3000-015) in OPTI-MEM media (Gibco, 31985062) for 15–30 min. The DNA-lipid complex was then added onto the cells in a drop-wise fashion. Cells were incubated with the siRNAs for at least 24 hr before harvesting for experimental use. All experiments were performed 24–72 hr post-transfection. All siRNA constructs were purchased from Dharmacon:

Individual siGENOME human TAOK1 siRNA, Dharmacon, (D-004846-02-0005)
Individual siGENOME human TAOK2 siRNA, Dharmacon, (D-004171-13-0005)
Individual siGENOME human TAOK3 siRNA, Dharmacon, (D-004844-02-0005) siGENOME Non-Targeting siRNA Pool #1, Dharmacon, (D-001206-13-05)

## Stable Lenti-viral cell line generation

293 T cells were plated in T-75 plates and were grown to ~70% confluence prior to transfection. 1 hr prior to transfection, the media was replaced with fresh media. Lentiviral vectors were incubated with third-generation viral packaging vectors (pCMV-VSV-G, pMDLg/pRRE, pRSV-Rev) and LT-1 (Mirus, MIR-2305) in OPTI-MEM media for 30 min as described by the provider's instructions. This transfection mix was then added onto the 293 T cells in a drop-wise fashion. Virus containing media was collected 24–48 hr after transfection. Cellular content and debris were removed by centrifugation, and the supernatant was filtered through 0.45 μm PES syringe (Millex, SLHP033RS) to remove any remaining cells in the media. Polybrene was added to the media at the final concentration of 8 ng/ml to facilitate viral transduction of the target cell line. The target cell lines were transduced with the virus-containing media for 24–48 hr. At the end of the transduction period, cells were washed,

trypsinized, and re-plated for selection. Transduced cells were exposed to high concentration antibiotic selection (3 µg/ml puromycin) up to 2 weeks (approximately three passages). All lentiviral procedures were carried out following Biosafety Level 3 (BSL3) practices in BSL2 tissue culture hoods according to institutional biosafety rules.

## Boyden chamber invasion assay

Each Boyden chamber membrane (Fisher Scientific, 353097) was coated with a thin layer of BME (200 µl of 0.25 mg/ml stock, or total of 50 µg of BME per membrane) and incubated at 37°C for 1 hr. Cells were trypsinized and centrifuged at 500 x g for 5 min followed by two rounds of PBS washes to remove remaining serum-containing media. Then, the cells were resuspended in serum-free media and diluted to the desired concentration for plating onto the Boyden chambers. Each Boyden chamber received 20,000–100,000 cells in 300 µl serum-free media, depending on the cell line. 10% serum was used as the chemoattractant for these assays. For the experiments testing the effect of MAPK inhibitors on invasion, the cells were resuspended in drug-containing serum-free media immediately. After 16–24 hr, the membranes were stained with Calcein AM (Fisher Scientific, 354217) for 1 hr at 37°C in the dark to stain for live cells. Cells that are in the top chamber were removed from the membrane with a wet cotton swab. Cells in the bottom chamber were dissociated from the membrane by incubating in cell dissociation buffer (Trevigen, Cultrex 3455-096-05) in a shaker at 37°C for 1 hr. Calcein AM signal was measured in Perkin Elmer Victor X3 plate reader as a read-out of invaded cells.

## High-throughput chemotactic invasion assays

For testing anti-invasive drug combinations, IncuCyte ClearView 96-Well Chemotaxis plates (Essen BioScience) were used. A total of 2000 cells per well were embedded in 2 mg/ml BME and plated onto the chemotaxis plate following the manufacturer's instructions. Media containing 2% FBS was used in both top and bottom chambers to maintain cell viability over 72 hr or more. 200 ng/ml human EGF (Bio-Techne, 236-EG-01M) was used as the chemotactic agent in the bottom chamber, and the control wells only had the vehicle for the chemotactic agent.

This assay is more accurate when nuclear-labeled cells are used. Therefore, we generated BM1-mKate2 (nuclear red) cells using IncuCyte NucLight Red Lentivirus Reagent (Essen BioScience, 4478) following the manufacturer's instructions. After transduction, cells with the highest nuclear red signal intensity (top 25%) were sorted by FACS.

The chemotaxis module in IncuCyte can accurately count the number of cells in the top chamber and the bottom chamber of the ClearView plates separately. Invasive capability of the cells in the presence of various small molecule inhibitors (SB203580, SP600125, Trametinib, URMC-099, CX-4945, SW-538) was measured as the percentage of cells that moved to the bottom chamber over the period of 72 hr. The formula used for this calculation is (number of cells in the bottom chamber)/ (number of cells in the bottom chamber +number of cells in the top chamber) x 100. The total number of cells in the top and bottom chambers is used as a readout of proliferation, which was important for determining drug combinations that blocked invasion without affecting growth properties of the cells.

## Proliferation assays

For proliferation assays, 1000–20,000 cells (depending on the cell line) were plated in 96-well plates and quantified over 3 days in IncuCyte by measuring confluence in Phase-Contrast images taken every 4 hr. For experiments testing the effect of MAPK inhibitors on proliferation, the cells were plated in 100 µl per well and allowed to adhere overnight. Then, 100 µl growth media containing 2X drug was added directly on top of the initial media.

## 3D cultures

For 3D proliferation experiments, we used Cultrex 3D Basement Membrane Matrix, Reduced Growth Factor (Trevigen, 3445-005-01, Lot No 37353J16, Lot concentration: 15.51 mg/ml, referred to as BME). For all experiments, the cells in growth media (2% FBS) were mixed with BME at a final concentration of 2 mg/ml. For 3D proliferation assays, 100 µl of the cell/BME mixture was dispensed into each well of a 96-well plate. Upon solidification of BME, 100 µl of growth media was added on

top of the solidified gel. For experiments where the cells were treated with inhibitors, the inhibitors were prepared in the growth media at 2X of their desired final concentration and added after the gel is solidified to assure 1X final concentration. The growth of the cells was monitored in IncuCyte Zoom or S3 models for the indicated duration of time.

## RNA isolation and qRT-PCR

Cells were washed with cold PBS twice and lysed in TRI Reagent (Zymo Research, R2050-1-200). RNA was isolated using Direct-zol RNA MiniPrep (Zymo Research, R2052). 4 μg of total RNA from each sample was converted to cDNA using High-Capacity cDNA Reverse Transcription Kit (Applied Biosystems, 4368813). Primer pairs used for this study are listed below.

List of Human primers used in this study:

*PEBP1* Forward: GCTCTACACCTTGGTCCTGACA
Reverse: AATCGGAGAGGACTGTGCCACT
*BACH1* Forward: CACCGAAGGAGACAGTGAATCC
Reverse: GCTGTTCTGGAGTAAGCTTGTGC
*NFATC2* Forward: GATAGTGGGCAACACCAAAGTCC
Reverse: TCTCGCCTTTCCGCAGCTCAAT
*ROCK1* Forward: GAAACAGTGTTCCATGCTAGACG
Reverse: GCCGCTTATTTGATTCCTGCTCC
*ROCK2* Forward: TGCGGTCACAACTCCAAGCCTT
Reverse: CGTACAGGCAATGAAAGCCATCC
*ADAM10* Forward: GAGGAGTGTACGTGTGCCAGTT
Reverse: GACCACTGAAGTGCCTACTCCA
*ADAM17* Forward: AACAGCGACTGCACGTTGAAGG
Reverse: CTGTGCAGTAGGACACGCCTTT
*EPC1* Forward: CCAGACATGCAGTACCTCTACG
Reverse: GCTGTTTCTGCATGAGTGCCAG
*PIKFYVE* Forward: CTGAGTGATGCTGTGTGGTCAAC
Reverse: CAAGGACTGACACAGGCACTAG
*DOCK4* Forward: GCATGTGGATGATTCCCTGCAG
Reverse: GGAGGTGATGTAACACGACAGG
*DOCK5* Forward: GCTTCTGAGCAACATCCTGGAG
Reverse: TCCTTCTCAGCAGCCGTTCCAT
*ARL13B* Forward: GAACCAGTGGTCTGGCTGAGTT
Reverse: GTTTCAGGTGGCAGCCATCACT
*DDR2* Forward: AACGAGAGTGCCACCAATGGCT
Reverse: ACTCACTGGCTTCAGAGCGGAA
*ITGA1* Forward: CCGAAGAGGTACTTGTTGCAGC
Reverse: GGCTTCCGTGAATGCCTCCTTT
*RAPGEF2* Forward: CTCGGATCAGTATCTTGCCACAG
Reverse: AGGTTCCACTGACAGGCAATGC
*RAPGEF6* Forward: AGACAGATGAGGAGAAGTTCCAG
Reverse: GACCTCATAGGCACTGGAGACA
*APC* Forward: AGGCTGCATGAGAGCACTTGTG
Reverse: CACACTTCCAACTTCTCGCAACG

List of Mouse primers used in this study:

*Pebp1* Forward: ACTCTACACCCTGGTCCTCACA
Reverse: TGAGAGGACAGTGCCACTGCTA
*Bach1* Forward: CCATGACATCCGCAGAAGGAGT
Reverse: GCGTTGACAGAATGTGGTCTCG
*Nfatc2* Forward: ACTTCACAGCGGAGTCCAAGGT
Reverse: GGATGTGCTTGTTCCGATACTCG
*Rock1* Forward: CACGCCTAACTGACAAGCACCA
Reverse: CAGGTCAACATCTAGCATGGAAC
*Rock2* Forward: GTGACCTCAAACAGTCTCAGCAG
Reverse: GACAACGCTTCTGAGTTTCCTGC
*Adam10* Forward: TGCACCTGTGCCAGCTCTGATG
Reverse: GATAGTCCGACCACTGAACTGC

*Adam17* Forward: TGTGAGCGGTGACCACGAGAAT
Reverse: TTCATCCACCCTGGAGTTGCCA
*Epc1* Forward: CTGCCAGGCTTCAGTGCTAAAG
Reverse: ACTGACAGCCTGCTTTCCTACG
*Pikfyve* Forward: TCTTCTGCCCAGTCCAGCAATG
Reverse: ACAGAACATGCTCGGACACTGG
*Dock4* Forward: GATAGGAGAGGTGGATGGCAAG
Reverse: CGCCTTGAGATGCAGATCGTAG
*Dock5* Forward: GAGCCGACAGTCTCCTCACATT
Reverse: CTGCCTGGTTTTGAAGGTGCTG
*Arl13b* Forward:ACCAGTGGTCTGGCTGAGATTG Reverse: CATCACTGTCCTTCTCCACGGT
*Ddr2* Forward: TCATCCTGTGGAGGCAGTTCTG
Reverse: CTGTTCACTTGGTGATGAGGAGC
*Itga1* Forward: GGCAGTGGCAAGACCATAAGGA
Reverse: CATCTCTCCGTGGATAGACTGG
*Rapgef2* Forward: GCCGAATGGCATCAGTCAACATG
Reverse: CAACATCCAGCACTGTGGCGTT
*Rapgef6* Forward: ACAGAGTGAGCCAGGTGCTTCA
Reverse: CACTCACTTCCTCAGTTGGTCC
*Apc* Forward: GTGGACTGTGAGATGTATGGGC
Reverse: CACAAGTGCTCTCATGCAGCCT

## Chromatin immunoprecipitation (ChIP)

BM1 cells were crosslinked with 1% formaldehyde for 10 min at 37°C and quenched with 125 mM glycine for 1 min. Cells were sheared in buffer including 0.1% SDS, 50 mM Tris-HCl (pH 7.6), I mM EDTA (pH8.0), 0.002% Triton X-100, supplemented with PMSF and protease inhibitor. Lysates were sonicated using Bioruptor in total four cycles (30 s on/30 s off each cycle). Sonicated chromatin was incubated with antibody (rabbit polyclonal anti-Bach1 (A1-6)) or normal rabbit serum conjugated with Dynabeads protein A/G for 150 mins at 4°C. Beads were washed twice for five mins each time with RIPA buffer (10 mM Tris-HCl pH7.6, 1 mM EDTA, 0.1% SDS, 0.1% NaDOC, 1% Triton X-100); RIPA buffer supplemented with 0.3M NaCl, LiCl buffer (0.21 M LiCl, 0.5% NP-40, 0.5% NaDOC) and TE buffer plus 0.2% Triton X-100. Next, beads were washed once with TE buffer for five mins. Beads were eluted in buffer including 0.003% SDS, 10 mM Tris-HCl (pH8.0) and 1 mM EDTA (pH 8.0), 0.1 mg/ml Proteinase K for 4 hr at 65°C. ChIP-DNA was purified by AMPure XP. To prepare for ChIP-seq, sonication step was optimized to 30 cycles and confirmed size chromatin fragments between 150 and 300 bp. ChIP-DNA samples were used to prepare DNA library and sequence as described (*Sato et al., 2020*). ChIP-DNA samples were used to prepare DNA library and sequence as described (*Sato et al., 2020*).

List of Human primers used for the ChIP assay quantitative RT-PCR:

*HMOX1* (positive cont.) Forward: AGTCGCGATTTCCTCATCCC
Reverse: TTCCCTTTGTTTCCGCGAGT
*ITGA1* Forward: GGTCTGAGTAACCCCACTTCC
Reverse: AGCACACCACAAAAGCCAAG
*DOCK4* Forward: ATTGTTGTGAAGGCCAACCC
Reverse: AGAAGGAGTGCAGTCTGGTTT
*RAPGEF2* Forward: GGGTGCTCCAATTGTATGTACTGAT
Reverse: TGATTCAGCTTTGGGGAGTGA
*PIKFYVE* Forward: CTGGACTCCTTCTGCCTGAG
Reverse: AAGACTCCGCCCTCTGTTTT
*ROCK2* Forward: GCATAGGAAGCGAGTACCCAT
Reverse: GACTCCTTTAGGCCCCGTCA
*RAPGEF6* Forward: CGCCACAGTTCATTCACACT
Reverse: GCGAAGGGTTGTTTGCTAGA
Negative cont. Forward: ATTTGCCTGGAGTGGAAGTG
Reverse: CTGTATCCAGGGGGATGATG

## Mouse studies

Mice were procured and housed by the Animal Resources Center and handled according to the Institutional Animal Care and Use Committee at the University of Chicago. Athymic nude mice were purchased from Harlan Sprague Dawley and C57Bl/6 mice were purchased from the Jackson Laboratories.

For primary tumor growth experiments, $2 \times 10^6$ BM1 cells or $5 \times 10^5$ LMB cells were injected orthotopically near the mammary fat pad of athymic nude or C57Bl/6 mice, respectively. Tumor growth was monitored over time by caliper measurements of the width and length of tumors. Tumor volumes were calculated with the formula:

$$volume = \frac{\pi}{6} \times width^2 \times length$$

The mice were sacrificed when the tumors reached approximately 1 cm$^3$.

For metastasis assays, $1 \times 10^5$ luciferase-expressing BM1 cells (BM1-luc) were injected into the left ventricle of the heart to allow for systemic distribution of the bone-tropic tumor cells. $5 \times 10^5$ LMB cells were injected into the tail vein. Mice were monitored for 3–6 weeks (depending on the model) for tumor development. At the earliest sign of respiratory problems or paralysis of the limbs, the experiment was ended, and the mice were euthanized. Tumor burden was measured at the end of the study via Xenogen IVIS 200 Imaging System (PerkinElmer) for BM1-luc tumors. For LMB tumors, tumor burden was measured by counting overt surface metastases in the lungs after perfusion and formalin fixation, as well as counting tumors in cross-sections of the lungs after H and E staining (described below).

For the in vivo studies involving MAPK inhibitors and the four-drug MAPKi combination treatment, small molecule inhibitors were resuspended under sterile conditions. Since not all of the inhibitors were water-soluble, all inhibitors were initially resuspended in DMSO at the volumes that will result in less than 5% final DMSO concentration. p38 inhibitor SB203580 and MLK inhibitor URMC-099 were further diluted to the desired concentration with 50 %PEG-400 (Sigma, 91893)+50 %saline. JNK inhibitor SP600125 and MEK inhibitor Trametinib were diluted in corn oil (Sigma, C8267). For the four-drug combinatorial treatment, all inhibitors were dissolved in their own solvent at 4X higher concentration then the desired final concentration. Then, SB203580 and URMC-099 were mixed at a 1:1 ratio, reducing the concentration for each drug down to 2X. Similarly, SP600125 and Trametinib were mixed at a 1:1 ratio. These dual combination solutions were then filtered through 0.22 μm PES filter syringes to assure sterility. Each mouse received 50 μl of each dual combination on the same day, resulting in a total of 100 μl of drug mix (two injections per mouse) with each drug at their desired 1X final concentration. Final concentration for SB203580, URMC-099, SP600125, or Trametinib in the four-drug MAPKi combination was 10 mg/kg/day, 10 mg/kg/day, 10 mg/kg/day, or 0.5 mg/kg/day, respectively. All injections were intraperitoneal.

For the tumor growth experiments with the MAPK inhibitors, tumors were allowed to reach the size 50–100 mm$^3$ size before the MAPKi treatment began. Then, the mice were treated with the respective MAPKi treatment (or the control) for up to 3 weeks. Tumor size was monitored twice a week with a caliper. For the metastasis assays, the tumor cells were treated with the four-drug MAPKi combination at the in vitro doses for 24 hr prior to injections to allow for anti-metastatic reprogramming of the cells. Homing to metastatic tissues upon intracardiac or tail vein injections can take up to 48 hr. To ensure that the reprogrammed tumor cells do not revert back to their untreated state in the circulation, we pre-treated the mice with the MAPKi combination 2–6 hr before tumor cell inoculation as well. After the inoculation, the mice were treated with the inhibitors daily for up to 3 weeks until the experimental endpoints discussed above were reached.

## Histology

Tumor tissues were fixed in 10% formalin upon dissection for 72 hr and then transferred into 70% Ethanol for long-term storage. Mouse lungs were perfused with PBS before formalin fixation step to allow for tissue expansion and high-quality histological analysis. Fixed tissues were embedded in paraffin and sliced into 5 μm sections prior to hematoxylin and eosin (H and E) staining. All tissue processing and staining for this body of work was performed by the University of Chicago Human Tissue Resource Center. For the detection of tissue morphology as well as tumor populations within the

lung, lung sections were deparaffinized, immersed in hematoxylin, rinsed in warm distilled water, and treated with eosin. Stained slides were scanned at 10X on a Nikon Eclipse Ti2 Inverted Microscope System.

## MIB-MS analysis

Multiplexed inhibitor beads – mass spectrometry analysis on BM1-VC and BM1-RKIP tumors was conducted as previously described (*Duncan et al., 2012*). Tumors were grown in athymic nude mice as described previously. Once the tumors reached the size of ~300 mm$^3$ they were isolated, flash-frozen in liquid nitrogen, and shipped to the Johnson Laboratories in Chapel Hill. Preparation of the lysate for the MIB-MS analysis, and the mass spectrometry were all performed as described by Duncan et al.

## RNA-sequencing

To compare the transcriptomes of metastatic BM1-VC and non-metastatic BM1-RKIP tumors, $1 \times 10^6$ cells were injected orthotopically. When tumors reached approximately 200 mm$^3$ size (about 3 weeks post inoculation), we harvested the tumors and flash-froze them in liquid nitrogen. Tumor samples were pulverized immediately, and lysed in TRI Reagent (Zymo Research, R2050-1-200). RNA was extracted using the Direct-zol RNA MiniPrep Kit (Zymo Research, R2052) following the manufacturer's instructions under RNAse-free conditions. In order to prevent contamination of the RNA samples by genomic DNA, the samples were treated with DNAse-I (Zymo Research, E1011-A) for 15 min at ambient temperature on the RNA extraction column. Total RNA was eluted in RNAse/DNAse-free water (Zymo Research, W1001-30) and submitted to the University of Chicago Genomics Facility for further analysis.

RNA quality assessment, library preparation, and sequencing of the tumor RNA samples were all performed by the Genomics Facility staff following the facility's standardized protocols. Quality of the samples were assessed using a Bioanalyzer, and the samples were determined to be of high quality with an average RNA integrity number (RIN) of 8.6. For the RNA-seq analysis, we had seven control tumors and five RKIP-overexpressing tumors, so we chose to generate an individual oligo dT selected, mRNA directional library for each tumor sample without any pooling scheme. All 12 samples were run on the same lane in HiSEQ4000 to generate 50 base-pair long single-end reads.

Bioinformatic analysis of the RNA-seq results were all carried out using the web-based bioinformatics platform Galaxy (usegalaxy.org). Raw '*.fastq' files were uploaded to the Galaxy servers via a file transfer protocol (FTP) software. The reads were analyzed for GC content using 'FastQC' and trimmed to remove adaptor sequences using 'Trim Galore!". The reads were mapped to the human genome (hg19) using RNA STAR. In all samples, 70–75% of the reads were uniquely mapped. The resulting '*.bam' files were used to count reads per gene with 'featureCounts'. Finally, read counts were normalized and analyzed for differential expression between Control and RKIP-overexpressing samples using 'DESeq2'. Principle component analysis on the normalized read counts demonstrated two distinct clusters of samples, separated by the RKIP status.

The raw and processed sequence data are deposited to Gene Expression Omnibus (GEO) with the series accession number GSE128983.

## The Cancer Genome Atlas (TCGA) analysis

For the analysis of patient data, normalized RNA-seq results were accessed through the cBioportal data base (http://www.cbioportal.org/) (*Gao et al., 2013*). For every TCGA cancer type, the provisional data sets were used for analysis (tagged 'TCGA, Provisional' on cBioportal). Lists of genes that correlate with RKIP (*PEBP1*) and *BACH1* were also downloaded directly from cBioportal, as the data base already has these correlation matrices generated for each TCGA cancer type. Oncotype and expression heatmap plots were directly generated by cBioportal. Prior to generation of these plots, z-score threshold of 0.5 was arbitrarily chosen to classify patients into high versus low expressors for a particular gene of interest. For example, if a patient's tumor sample has an RKIP expression level that has a z-score higher than 0.5, then the sample was deemed 'RKIP-high', and if the z-score was below −0.5, the sample was deemed 'RKIP-low'. If the z-score falls within −0.5 and 0.5, then the sample was considered as 'Intermediate', or 'Other'. Both Pearson and Spearman

correlations were used in determining gene-gene correlations and a coefficient cut-off of 0.3 was chosen arbitrarily for both correlation metrics.

## Gene set enrichment analyses

Functional gene set enrichment analysis of the differentially expressed genes in the RNA-seq data as well as the genes that correlate with RKIP (*PEBP1*) and *BACH1* was performed using the web-based interface of the Metascape software (metascape.org) (*Tripathi et al., 2015*). For the identification of pathways and processes enriched in the input gene lists, both 'Gene Ontology' (GO) and 'Kyoto Encyclopedia of Genes and Genomes' (KEGG) categories were considered. A minimum overlap of five genes and an enrichment score of 1.5 were chosen as the enrichment parameters. An adjusted p-value cut-off of 0.05 was chosen as the significance threshold.

## Network model

### Model description

We propose a coarse-grained framework for devising a new cancer treatment strategy based on cellular reprogramming. The networks governing the cell dynamics involve a plethora of components interacting in a non-linear fashion, and its quantitative description would require, in principle, the construction of a large system of coupled differential equations. That approach is unfeasible because of a lack of detailed knowledge of the parameters and chemical reactions that they govern. Hence, an alternative approach was used to describe the signal flow within the network which drives the transcription of *BACH1* by MAPKs under stress assuming a steady state regime.

We consider that a cancer cell has a multiplicity of hierarchically structured driver networks responsible for activation of the stress response genes that promote metastasis. The BACH1 driver network is a primary absorber of the stress signal that, when compromised, may generate a surplus of kinase signal(s) to activate secondary compensatory networks, enabling redundancy of stress processing and response. Our data indicate that the BACH1 driver network is fully operational in all cell lines and that the stress signal strength can induce saturation of the activity of all components of the network.

We model the driver network considering the stress signal as a steady state flow through the network pathways and the kinase nodes. The nodes of the network are labeled accordingly with the kinase that we posit they represent. The inflow of a node $i$ indicates the arrival of the kinase signal(s) into it. The inflow interacts with the kinase node, and the outflow indicates the product of this interaction. Each node has an activity that regulates the inflow of kinase signal(s), and at each node the outflow of products is equal to the inflow of the signal.

The treatment targeting a specific node of the network will reduce its activity and, hence, its capacity for consuming the products (signal) from its upstream node. The surplus of products from the upstream nodes may be either redirected within the network or, in the case of saturation of the crosstalk links, be redirected to activate a secondary compensatory network. In the case of the surplus being greater than the threshold of activation of the compensatory network, an alternative system (driver network) will be turned on and the stress response will still be functional and capable of promoting metastasis.

## Signal flow description

*Figure 5A* shows a phenomenological representation of the *BACH1* transcription driver network N1 for the BM1 cell line. The nodes containing a kinase are labeled by an index $i = 2, \ldots, 8$BACH1 while node 1 indicates the splitting of the stress signal and node 9 denotes the induction of *BACH1* transcription as generated by the inflow of upstream kinase signals. The inflow of the $i$BACH1 -th node is denoted by $\Upsilon_i$BACH1 while its outflow is $\Omega_i$BACH1. Since we are considering a conservative network, we have $\Omega_i = \Upsilon_i$BACH1, for $i = 1, \ldots, 9$BACH1. The flow capacity of arrow connecting node $i$BACH1 to $j$BACH1 is indicated by $c_{ij}$BACH1 which is a fraction of $\Upsilon_i$BACH1 when the node has more than one outgoing pathway. Hence, a coefficient $\alpha_i$BACH1 or $A_i$BACH1 besides a pathway denotes the fraction of the inflow going through it such that $\alpha_1 + \alpha_2 + \alpha_{12} = 1$BACH1, and $\alpha_k + A_k = 1$BACH1 where $k = 3, \ldots, 6$BACH1. The absence of those coefficients indicate that the outflow equals the inflow. The intensity of the stress signal is denoted by $\Sigma$BACH1 and the degree of activation of transcription of *BACH1* caused by this signal is indicated by $\phi_{\text{BACH1}}$BACH1. For the

case of maximal stress signaling, we have $\phi_{\mathrm{BACH1}} = \Sigma\mathrm{BACH1}$. The treatment generates a surplus of products of the functional network denoted by $E\mathrm{BACH1}$ such that the degree of activation of transcription of *BACH1* under treatment is $\phi_{\mathrm{BACH1}} = \Sigma - E\mathrm{BACH1}$. The degree of activation of transcription of *BACH1* can be rewritten as a fraction of maximal stress; $\phi_{\mathrm{BACH1}} = 1 - \epsilon\mathrm{BACH1}$. For simplicity we only represent treatment targeting MEK and its effect on generating RAF surplus.

Let us write the formulae for the input flows in fractions of the stress signal $\Sigma$.

We have an input signal being split at node 1 such that $\Upsilon_1 = 1$ and, since node 1 has three outgoing pathways, we have $\Omega_1 = \Upsilon_1 = \alpha_1 + \alpha_2 + A_{12} = 1$ which implies on: $\Upsilon_2 = \alpha_1; \Upsilon_5 = \alpha_2; \Upsilon_7 = A_{12}$.

Node 2 has one outgoing pathway and generates an outflow $\Omega_2 = \Upsilon_2$ that fully inflows node 3, hence $\Upsilon_3 = \Omega_2 = \Omega_3$. Node 5 has two outgoing pathways, hence $\Omega_5 = \Upsilon_5 = \alpha_3\Upsilon_5 + A_3\Upsilon_5 = \alpha_2\alpha_3 + \alpha_2 A_3$ and, similarly, node 7 outflow obeys $\Omega_7 = \Upsilon_7 = A_{12}\alpha_4 + A_{12}A_4$.

The inflow of node 8 is $\Upsilon_8 = A_3\Omega_5 + A_4\Omega_7$ such that $\Omega_8 = \Upsilon_8$; the inflow of node 4 becomes $\Upsilon_4 = A_5\Omega_8 + \Omega_3$ such that $\Omega_4 = \Upsilon_4$; and the inflow of node 6 is $\Upsilon_6 = A_6\Omega_4 + \alpha_3\Omega_5 + \alpha_4\Omega_7$ such that $\Omega_6 = \Upsilon_6$ ; $\Omega_3, \Omega_5$, and $\Omega_7$ are defined above.

The inflow arriving at node 9 is written as $\Upsilon_9 = \alpha_6\Omega_4 + \alpha_5\Omega_8 + \Omega_6$ such that in the absence of treatment, $\phi_{\mathrm{BACH1}} = \Upsilon_9 = 1$ as it can be verified by direct substitution. Indeed, let us consider $\Omega_8 = A_3\Omega_5 + A_4\Omega_7$, $\Omega_4 = A_5\Omega_8 + \Omega_3$, and $\Omega_6 = A_6\Omega_4 + \alpha_3\Omega_5 + \alpha_4\Omega_7$, such that $\Upsilon_9$ becomes $\Upsilon_9 = (\alpha_6 + A_6)\Omega_3 + ((\alpha_6 A_5 + A_6 A_5)A_3 + \alpha_5 A_3 + \alpha_3)\Omega_5 + ((\alpha_6 A_5 + A_6 A_5)A_4 + \alpha_5 A_4 + \alpha_4)\Omega_7$. Since $\alpha_k + A_k = 1$ for $k = 3, \ldots, 6$, we obtain $\Upsilon_9 = \Omega_3 + \Omega_5 + \Omega_7 = \Upsilon_2 + \Upsilon_3 + \Upsilon_7 = 1$.

The parameters governing the flow of information through the network were chosen based upon experimental results, such that the values of the parameters $\alpha_k$ indicate the strength of absorption of the kinase from the upstream node by the current one. For N1, we have:

$$\alpha_1 = 0.2$$

and the complementary flow parameters are $A_{12} = 1 - \alpha_1 - \alpha_2$ and $A_k = 1 - \alpha_k$ for $k = 3, \ldots, 6$.

For N2, we eliminate the crosstalk and the parameters become:

$$\alpha_3 = 1$$

so that $A_4 = 1$ and $A_3 = A_5 = A_6 = 0$.

## BACH1 transcription after treatment

We test our approach by analyzing four treatment scenarios using (1) p38i, (2) MEKi, (3) JNKi, and (4) 4D-MAPKi. The reduction in target kinase activity caused by those drugs will be proportional to a function of the drug activity reduction denoted as $\xi_{\mathrm{p38}}$, $\xi_{\mathrm{MEK}}$, $\xi_{\mathrm{JNK}}$, and $\xi_{\mathrm{4D-MAPK}} = \xi_{\mathrm{p38}} + \xi_{\mathrm{MEK}} + \xi_{\mathrm{JNK}} + \xi_{\mathrm{MLK}}$. That reduction will cause a surplus of kinase signals from the upstream nodes that is proportional to the reduction in the activity of the target node. Hence, the inflow of the target node will be given by $1 - \xi_{\mathrm{p38}}\Upsilon_8$, $1 - \xi_{\mathrm{MEK}}\Upsilon_3$, $1 - \xi_{\mathrm{JNK}}\Upsilon_6$, and $1 - (\xi_{\mathrm{p38}}\Upsilon_8 + \xi_{\mathrm{MEK}}\Upsilon_3 + \xi_{\mathrm{JNK}}\Upsilon_6 + \xi_{\mathrm{MLK}}\Upsilon_6)$, where $\Upsilon_i$ is the inflow to the target without treatment. The total surplus generated by each single drug treatment is $\xi_{\mathrm{p38}}\Upsilon_8$, $\xi_{\mathrm{MEK}}\Upsilon_3$, $\xi_{\mathrm{JNK}}\Upsilon_6$, while for the 4D-MAPKi we assume additive effect as a first approximation which results in $\xi_{\mathrm{p38}}\Upsilon_8 + \xi_{\mathrm{MEK}}\Upsilon_3 + \xi_{\mathrm{JNK}}\Upsilon_6 + \xi_{\mathrm{MLK}}\Upsilon_6$.

We will indicate the fraction of inhibition caused by a given treatment dose by $x\mathrm{BACH1}$, since the drug dosage may vary with its specific function. For the drugs targeting nodes p38, MEK, and MLK, we can represent the reduction in *BACH1* transcription effectively as a function of a dose $x\mathrm{BACH1}$ targeting the $i\mathrm{BACH1}$-th node:

$$\phi_{\mathrm{BACH1}}(x) = \frac{1}{1 + K_i x},$$

where $K_i\mathrm{BACH1}$ is a constant set according to the effect of treatment on reducing *BACH1* transcription. The latter is assumed to be equal to the inflow $\Upsilon_9\mathrm{BACH1}$. Then, *BACH1* relative transcription after reduction of the activity of node $i\mathrm{BACH1}$ by a fraction $\xi_i\Upsilon_i\mathrm{BACH1}$ can be written as:

$$\phi_{\mathrm{BACH1}}(x) = 1 - \xi_i\Upsilon_i,$$

where $i$ denotes the node or its corresponding kinase. Hence, we can write the function for the reduction of the node activity as a consequence of drug action as:

$$\xi_i(x) = \frac{\Delta_i x}{1 + K_i x},$$

where $\Delta_i = K_i / \Upsilon_i$, and $\Upsilon_i$ is the inflow at node $i$ without treatment.

## Treatment with p38i

*BACH1* transcription after this treatment can be approximated for $K_1 = 2$, such that:

$$\phi_{\text{BACH1}}^{\text{p38i}}(x) = \frac{1}{1 + K_1 x}.$$

## Treatment with MEKi

*BACH1* transcription after this treatment can be approximated for $K_2 = 0.4$, such that:

$$\phi_{\text{BACH1}}^{\text{MEKi}}(x) = \frac{1}{1 + K_2 x}.$$

## Treatment with MLKi

*BACH1* transcription after this treatment can be approximated for $K_4 = 0.1$, such that:

$$\phi_{\text{BACH1}}^{\text{MEKi}}(x) = \frac{1}{1 + K_4 x}.$$

## Treatment with JNKi

The output of *BACH1* after this treatment is obtained using a different model because JNK is repressing node 4 and has a compensatory crosstalk with node 8. Hence the activity of those nodes will increase because of reduction of activity of node 6. Let us describe the reduction of the activity of node 4 as

$$\Upsilon_4' = \Upsilon_4 \left( 1 + \frac{4}{1 + (\gamma_1 \Upsilon_7)^4} \frac{1}{1 + (\gamma_2 \Upsilon_6')^4} \right),$$

where the primed symbols indicate the inflows under treatment. Since node 4 is also repressed by node 7, and the network is operating under maximal stress, we assume that node 7 represses node 4 maximally such that node 4 activity remains almost constant even under reduction of the activity of node 6. This is based on assuming that the capacity of the pathways connecting nodes 3 and 8 to node 4 will not be affected by a reduction in activity of node 6, and, hence: $\Upsilon_4' = \Upsilon_4 = c_{34} + c_{84} = \Omega_3 + A_5 \Omega_8$.

However, the activity of node 8 will be affected by treatment inactivation of JNK redirecting the flow from nodes 5 and 7. Let us assume that the inflow of node 8 coming from nodes 5 and 7 is, respectively, determined by the activity of node 6 accordingly with

$$A_3 = \frac{1}{1 + (\gamma_3 \Upsilon_6(x))^4}, \text{ and } A_4 = \frac{1}{1 + (\gamma_4 \Upsilon_6(x))^4},$$

where $\gamma_3$BACH1 and $\gamma_4$BACH1 are two arbitrary constants and the activity of node 6 can be written as a function of the drug dosage $x$BACH1 targeting it. The treatment targeting node 6 reduces its activity and induces a compensatory flow to pass through node 8 which is assumed to be capable of absorbing only two thirds of the surplus of kinase signals coming from nodes 5 and 7. Therefore, the other one-third of surplus will be redirected toward two compensatory networks, each of them receiving the surplus of one of the kinase signals. Then, the surplus is the difference between the absorption by node 8 without and with treatment, and *BACH1* transcription becomes

$$\phi_{\text{BACH1}} = 1 + \frac{1}{3} \left( \frac{1}{1 + (\gamma_3 \Upsilon_6(x))^4} + \frac{1}{1 + (\gamma_4 \Upsilon_6(x))^4} - \frac{1}{1 + (\gamma_3 \Upsilon_6(0))^4} - \frac{1}{1 + (\gamma_4 \Upsilon_6(0))^4} \right),$$

where the factor 13 occurs because we are assuming that node 8 absorbs only two-thirds of the surplus generated by treatment. The inactivation of node 6 by treatment can be described by a function

$$\Upsilon_6(x) = \frac{\Upsilon_6(0)}{1 + K_3 x},$$

where $K_3 = 0.475628 \text{BACH1}$ is an arbitrary constant. We also set the values of the constants $\gamma_3 = 1.520908 \text{BACH1}$ and $\gamma_4 = 1.435780 \text{BACH1}$ arbitrarily. Our choice enables us to estimate the node activity under treatment as a fraction of its activity without treatment and to obtain a qualitative description of experimental data based on the reduction of *BACH1* transcription following treatment. The analysis of *BACH1* transcription after treatment is shown in for N1 and N2, respectively.

## Surplus analysis

The surplus for each treatment scenario can be evaluated using the output functions after treatment considering that the reduction of activity of a given kinase generates a surplus that comprises the non-absorbed quantities of kinase signal produced at the upstream nodes.

### Surplus of treatment with MEKi

The total surplus of RAF generated by this treatment can be computed from the transcription of BACH1. The surplus of RAF as function of the treatment is denoted by $\epsilon_{\text{RAF}}(x)$, such that $\epsilon_{\text{RAF}}(x) = \xi_{\text{MEK}}(x)\Upsilon_3$. Since $\xi_{\text{MEK}}(x) = \frac{\Delta_2 x}{1 + K_2 x}$, and $\Delta_2 = \frac{K_2}{\Upsilon_3} = \frac{K_2}{\alpha_1}$, the surplus of RAF is:

$$\epsilon_{\text{RAF}}(x) = \frac{K_2 x}{1 + K_2 x}.$$

The RAF surplus from MEKi is depicted in Graph 6B, and the function describing it is the same for the 4D-MAPKi treatment. The validity of the formula for both treatment scenarios is because we are assuming that the nodes of the network are operating at their maximal capacity and the surplus is not redirected within the network unless there is a repressive interaction.

### Surplus of treatment with MLKi

This treatment generates a surplus of stress signal which we assume not being redirected within the driver network. Therefore, this signal will be redirected to a compensatory network and its amount is denoted by $\epsilon_{\text{SIG}}(x)$, such that $\epsilon_{\text{SIG}}(x) = \xi_{\text{MLK}}(x)\Upsilon_5$. Since $\xi_{\text{MLK}}(x) = \frac{\Delta_4 x}{1 + K_4 x}$, and $\Delta_4 = \frac{K_4}{\Upsilon_5} = \frac{K_4}{\alpha_2}$, the surplus of stress signal is given by:

$$\epsilon_{\text{SIG}(x)} = \frac{K_4 x}{1 + K_4 x}.$$

The signal surplus is shown in Graph 6C for the 4D-MAPKi treatment.

### Surplus of treatment with JNKi and p38i

We compute the surplus generated at nodes 5 and 7 by treatments with JNKi and p38i based on the total surplus that they generate.

### Total surplus of treatment with JNKi

The surplus generated by this treatment is given by one third of the difference between the flow toward node 8 without and with treatment as previously considered when evaluating *BACH1* transcription under this treatment. We denote the surplus generated by treatment targeting node 6 as $\epsilon_6(x)\text{BACH1}$ which becomes

$$\epsilon_6 = \frac{1}{3}\left(\frac{1}{1 + (\gamma_3 \Upsilon_6(x))^4} + \frac{1}{1 + (\gamma_4 \Upsilon_6(x))^4} - \frac{1}{1 + (\gamma_3 \Upsilon_6(0))^4} - \frac{1}{1 + (\gamma_4 \Upsilon_6(0))^4}\right).$$

Note, however, that the outflow from node 5 is also affected by treatment by means of its inactivation by drug MLKi. Hence, we have $\alpha_2 \to \alpha_2/(1 + K_4 x)$ and the inflow of node 6 becomes

$$\Upsilon_6(x) = \frac{1}{1+K_3 x}\left\{\frac{\alpha_2}{1+K_4 x}\alpha_3 + A_{12}\alpha_4 + \left[\alpha_1 + \left(\frac{\alpha_2}{1+K_4 x}A_3 + A_{12}A_4\right)A_5\right]A_6\right\},$$

where we are rebalancing the inflow of node 6 without treatment, indicated within curly brackets, by its inactivation because of treatment, which is denoted by the term outside the curly brackets. The term within the curly brackets also has the node 5 outflow reduction because of treatment.

## Total surplus with treatment with p38i

The surplus generated by this treatment is given by the sum of the inflow toward node 8 without treatment plus the redirected flow from node 6 balanced by the reduction of activity of nodes 5 and 8 because of treatment. We can denote the surplus generated by treatment targeting node 8 by $\epsilon_8(x)$ such that

$$\epsilon_8(x) = \frac{K_1 x}{1+K_1 x},$$

where this surplus is the result of the combination of the surplus signals generated by treatment targeting node 6 and node 8 itself.

The surplus from node 5 can be computed from the sum of the fraction of surplus generated by treatment targeting nodes 6 and 8 where those fractions are proportional to the absorption capacity of each pathway coming from node 5 to nodes 6 or 8. We denote the surplus of node 5 by $\epsilon_{\text{MLK}}$ such that:

$\epsilon_{\text{MLK}} = \epsilon_6(x)\Gamma_3(x) + \alpha_4\epsilon_8(x)$, where,

$$\Gamma_3(x) = \frac{1}{1+(\gamma_3\Upsilon_6(x))^4}.$$

We proceed analogously to compute the surplus of node 7, denoted by $\epsilon_{\text{TAOK}}$, and obtain:

$$\epsilon_{\text{TAOK}} = \epsilon_6(x)(1-\Gamma_3(x)) + A_4\epsilon_8(x).$$

# Acknowledgements

The results shown here are in whole or part based upon data generated by the TCGA Research Network: https://www.cancer.gov/tcga. We thank Ani Solanki for his technical assistance with the animal experiments, and Kazuhiko Igarashi and Mitsuyo Matsumoto for providing the anti-BACH1 antibody (AI-6) for ChIP analyses as well as processing the ChIP-seq data. We also thank members of the Rosner Laboratory, Gabor Balazsi, and Robert Rosner for helpful comments.

# Additional information

## Competing interests

Marsha R Rosner: This research is also the subject of a pending US patent application # 17/048,282. The other authors declare that no competing interests exist.

## Funding

| Funder | Grant reference number | Author |
| --- | --- | --- |
| National Institutes of Health | R01 GM121735-01 | Marsha R Rosner |
| National Institutes of Health | CA058223 | Gary L Johnson |
| University of Chicago | Rustandy Fund | Marsha R Rosner |
| University of Chicago | Women's Board Grants Fund | Ali Ekrem Yesilkanal |
| University of Sao Paulo | Use of Intelligent Systems | Alexandre F Ramos |
| University of São Paulo | 18.5.245.86.7 | Alexandre F Ramos |

| Coordenação de Aperfeiçoamento de Pessoal de Nível Superior | 88881.062174/2014-01 | Alexandre F Ramos |
| Coordenação de Aperfeiçoamento de Pessoal de Nível Superior | | Alan U Sabino |

The funders had no role in study design, data collection and interpretation, or the decision to submit the work for publication.

### Author contributions

Ali Ekrem Yesilkanal, Conceptualization, Formal analysis, Investigation, Visualization, Methodology, Writing - original draft, Project administration, Writing - review and editing; Dongbo Yang, Payal Tiwari, Long Chi Nguyen, Xiao-He Xie, Siqi Sun, Lydia Robinson-Mailman, Ethan Steinberg, Timothy Stuhlmiller, Casey Frankenberger, Investigation; Andrea Valdespino, Christopher Dann, Validation, Investigation; Alan U Sabino, Investigation, Methodology; Jiyoung Lee, Investigation, Writing - review and editing; Elizabeth Goldsmith, Resources; Gary L Johnson, Funding acquisition, Investigation, Methodology, Writing - review and editing; Alexandre F Ramos, Supervision, Funding acquisition, Investigation, Methodology, Writing - original draft, Writing - review and editing; Marsha R Rosner, Conceptualization, Supervision, Funding acquisition, Methodology, Writing - original draft, Project administration, Writing - review and editing

### Author ORCIDs

Ali Ekrem Yesilkanal (iD) https://orcid.org/0000-0003-4988-1294
Alan U Sabino (iD) http://orcid.org/0000-0003-1094-5078
Jiyoung Lee (iD) http://orcid.org/0000-0001-8503-4805
Gary L Johnson (iD) http://orcid.org/0000-0003-2867-0551
Marsha R Rosner (iD) https://orcid.org/0000-0001-6586-8335

### Ethics

Animal experimentation: All animal protocols related to mouse experiments were approved by the University of Chicago Institutional Animal Care and Use Committee (IACUC #72228).

### Decision letter and Author response

Decision letter https://doi.org/10.7554/eLife.59696.sa1
Author response https://doi.org/10.7554/eLife.59696.sa2

## Additional files

### Supplementary files

• Transparent reporting form

### Data availability

RNA sequencing data have been deposited in GEO under the accession code GSE128983.

The following dataset was generated:

| Author(s) | Year | Dataset title | Dataset URL | Database and Identifier |
|---|---|---|---|---|
| Yesilkanal AE, Rosner M | 2019 | Gene expression data from BM1 (1833) tumors that are wild type or overexpressing RKIP in a xenograft mouse model | https://www.ncbi.nlm.nih.gov/geo/query/acc.cgi?acc=GSE128983 | NCBI Gene Expression Omnibus, GSE128983 |

The following previously published dataset was used:

| Author(s) | Year | Dataset title | Dataset URL | Database and Identifier |
|---|---|---|---|---|
| Gao J, Aksoy BM, Dogrusoz U, Dresdner G, Gross B, Sumer SO, Sun Y, Jacobsen A, Sinha R, Larsson E, Cerami E, Sander C, Schultz N | 2013 | The Cancer Genome Atlas (TCGA) | https://www.cbioportal.org/ | cBioPortal, The Cancer Genome Atlas (TCGA Firehose Legacy), www.cbioportal.org/ |

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

# Appendix 1

**Appendix 1—key resources table**

| Reagent type (species) or resource | Designation | Source or reference | Identifiers | Additional information |
|---|---|---|---|---|
| cell line (*Homo sapien*) | BM1 (triple negative breast cancer) | *Kang et al., 2003* | RRID:CVCL_DP48 | Derived from MDA-MB-231 cell line |
| cell line (*Homo sapien*) | MB436 (triple negative breast cancer) | ATCC | ATCC Cat# HTB-130, RRID:CVCL_0623 | |
| cell line (*Homo sapien*) | MCF10A; 184A1 (normal mammary epithelial) | ATCC | ATCC Cat# CRL-10317, RRID:CVCL_0598; ATCC Cat# CRL-8798, RRID:CVCL_3040 | |
| cell line (*Homo sapien*) | 293T (embryonic kidney) | ATCC | ATCC Cat# CRL-3216, RRID:CVCL_0063 | |
| cell line (*M. musculus*) | LMB (mouse triple negative breast cancer) | *Johnstone et al., 2015* | | |
| cell line (*M. musculus*) | M6C (mouse triple negative breast cancer) | *Holzer et al., 2003* | RRID:CVCL_A4AV | |
| chemical compound, drug | SP600125 | APExBIO | A4604 | JNK inhibitor |
| chemical compound, drug | Trametinib (GSK1120212) | APExBIO | A3018 | MEK inhibitor |
| chemical compound, drug | URMC-099 | APExBIO | B4877 | MLK inhibitor |
| chemical compound, drug | SB203580; SB203580-HCl | Selleckchem; APExBIO | S1076; B1285 | p38 inhibitor; water soluble p38 inhibitor |
| chemical compound, drug | CX-4945 (Silmitasertib) | APExBIO | A833010 | CK2 inhibitor |
| chemical compound, drug | SW-538 (SW034538) | *Piala et al., 2016* | | TAOK inhibitor |
| chemical compound, drug | anisomycin | Sigma-Aldrich | A9789 | |
| other | Odyssey Blocking Buffer | LI-COR Biosciences | 927–40010 | diluted 1:1 with PBS |
| software, algorithm | Image Studio Lite | LI-COR Biosciences | RRID:SCR_013715 | |
| other | Pierce ECL Western Blotting Substrate | Thermo Scientific | 32106 | |

*Continued on next page*

*Appendix 1—key resources table continued*

| Reagent type (species) or resource | Designation | Source or reference | Identifiers | Additional information |
|---|---|---|---|---|
| antibody | Anti-Phospho-TAOK3 (Ser177) + Phospho TAOK2 (Ser181) + Phospho-TAOK1 (Ser181) (Rabbit monoclonal) | Abcam | Abcam Cat# ab124841, RRID:AB_10974224 | WB (1:1000) |
| antibody | Anti-Phopsho-p44/42 MAPK(ERK1/2) (Thr202/Tyr204) (Rabbit polyclonal) | Cell Signaling | Cell Signaling Technology Cat# 9101, RRID:AB_331646 | WB (1:2000) |
| antibody | Anti-Phospho-SAPK/JNK (Thr183/Tyr185) (Rabbit polyclonal) | Cell Signaling | Cell Signaling Technology Cat# 9251, RRID:AB_331659 | WB (1:1000) |
| antibody | Anti-Phospho-p38 MAPK (Thr180/Tyr182) (Rabbit monoclonal) | Cell Signaling | Cell Signaling Technology Cat# 4511, RRID:AB_2139682 | WB (1:1000) |
| antibody | Anti-Phospho-AKT1 (S473) (Rabbit monoclonal) | Cell Signaling | Cell Signaling Technology Cat# 4060, RRID:AB_2315049 | WB (1:1000) |
| antibody | Anti-alpha-Tubulin (mouse monoclonal, IgM) | Santa Cruz | Santa Cruz Biotechnology Cat# sc-8035, RRID:AB_628408 | WB (1:1000-1:8000) |
| antibody | Anti-alpha-Tubulin (mouse monoclonal, IgM) | Invitrogen | Thermo Fisher Scientific Cat# MA1-19401, RRID:AB_2210198 | WB (1:4000) |
| antibody | Anti-GAPDH | Santa Cruz | Santa Cruz Biotechnology Cat# sc-32233, RRID:AB_627679 | WB (1:4000) |
| antibody | Anti-BACH1 | Santa Cruz | Santa Cruz Biotechnology Cat# sc-271211, RRID:AB_10608972 | WB (1:1000) |
| transfected construct (human) | si-TAOK1 (Individual siGENOME human TAOK1 siRNA) | Dharmacon | D-004846-02-0005 | |
| transfected construct (human) | si-TAOK2 (Individual siGENOME human TAOK2 siRNA) | Dharmacon | D-004171-13-0005 | |
| transfected construct (human) | si-TAOK3 (Individual siGENOME human TAOK3 siRNA) | Dharmacon | D-004844-02-0005 | |
| sequence-based reagent | si-NT (siGENOME Non-Targeting siRNA Pool #1) | Dharmacon | D-001206-13-05 | |
| recombinant DNA reagent | pCDH-EF1 (plasmid) | Addgene | RRID:Addgene_72266 | Lentiviral empty vector control |
| recombinant DNA reagent | pCDH-EF1-RKIP (plasmid) | *Dangi-Garimella et al., 2009* | | Lentiviral vector overexpressing RKIP |

*Continued on next page*

*Appendix 1—key resources table continued*

| Reagent type (species) or resource | Designation | Source or reference | Identifiers | Additional information |
|---|---|---|---|---|
| other | Calcein AM | Fisher Scientific | 354217 | Live cell marker |
| other | IncuCyte ClearView 96-Well Chemotaxis plates | Essen Biosciences | 4582 | High-throughput invasion assay platform |
| recombinant DNA reagent | IncuCyte NucLight Red Lentivirus Reagent | Essen Biosciences | 4478 | Lentiviral vectror containing mKate2 nuclear dye |
| other | BME (3-D Culture Matrix Reduced Growth Factor Basement Membrane Extract, PathClear) | Trevigen | 3445-005-01 | Lot No 37353J16, lot concentration: 15.51 mg/ml |
| peptide, recombinant protein | EGF (human) | Bio-Techne | 236-EG-01M | |
| software, algorithm | Chemotaxis module for IncuCyte Zoom or S3 | Essen Biosciences | Essen Incucyte Incucyte, RRID:SCR_019874; IncuCyte Chemotaxis Software, RRID:SCR_017316 | Special software module to analyse high-throughput invasion assays |
| sequence-based reagent | Hs_PEBP1 forward (*Homo sapiens*) | This paper | qRT-PCR primers | GCTCTACACCTTGGTCC TGACA |
| sequence-based reagent | Hs_PEBP1 reverse (*Homo sapiens*) | This paper | qRT-PCR primers | AATCGGAGAGGACTG TGCCACT |
| sequence-based reagent | Hs_BACH1 forward (*Homo sapiens*) | This paper | qRT-PCR primers | CACCGAAGGAGACAGTGAA TCC |
| sequence-based reagent | Hs_BACH1 reverse (*Homo sapiens*) | This paper | qRT-PCR primers | GCTGTTCTGGAGTAAGCTTG TGC |
| sequence-based reagent | Hs_NFATC2 forward (*Homo sapiens*) | This paper | qRT-PCR primers | GATAGTGGGCAACACCAAAG TCC |
| sequence-based reagent | Hs_NFATC2 reverse (*Homo sapiens*) | This paper | qRT-PCR primers | TCTCGCCTTTCCGCAGC TCAAT |
| sequence-based reagent | Hs_ROCK1 forward (*Homo sapiens*) | This paper | qRT-PCR primers | GAAACAGTGTTCCATGC TAGACG |
| sequence-based reagent | Hs_ROCK1 reverse (*Homo sapiens*) | This paper | qRT-PCR primers | GCCGCTTATTTGATTCCTGC TCC |
| sequence-based reagent | Hs_ROCK2 forward (*Homo sapiens*) | This paper | qRT-PCR primers | TGCGGTCACAACTCCAAGCC TT |
| sequence-based reagent | Hs_ROCK2 reverse (*Homo sapiens*) | This paper | qRT-PCR primers | CGTACAGGCAATGAAAGCCA TCC |
| sequence-based reagent | Hs_ADAM10 forward (*Homo sapiens*) | This paper | qRT-PCR primers | GAGGAGTGTACGTG TGCCAGTT |

*Continued on next page*

*Appendix 1—key resources table continued*

| Reagent type (species) or resource | Designation | Source or reference | Identifiers | Additional information |
|---|---|---|---|---|
| sequence-based reagent | Hs_ADAM10 reverse (*Homo sapiens*) | This paper | qRT-PCR primers | GACCACTGAAGTGCCTACTCCA |
| sequence-based reagent | Hs_ADAM17forward (*Homo sapiens*) | This paper | qRT-PCR primers | AACAGCGACTGCACGTTGAAGG |
| sequence-based reagent | Hs_ADAM17 reverse (*Homo sapiens*) | This paper | qRT-PCR primers | CTGTGCAGTAGGACACGCCTTT |
| sequence-based reagent | Hs_EPC1 forward (*Homo sapiens*) | This paper | qRT-PCR primers | CCAGACATGCAGTACCTCTACG |
| sequence-based reagent | Hs_EPC1 reverse (*Homo sapiens*) | This paper | qRT-PCR primers | GCTGTTTCTGCATGAGTGCCAG |
| sequence-based reagent | Hs_PIKFYVE forward (*Homo sapiens*) | This paper | qRT-PCR primers | CTGAGTGATGCTGTGTGGTCAAC |
| sequence-based reagent | Hs_PIKFYVE reverse (*Homo sapiens*) | This paper | qRT-PCR primers | CAAGGACTGACACAGGCACTAG |
| sequence-based reagent | Hs_DOCK4 forward (*Homo sapiens*) | This paper | qRT-PCR primers | GCATGTGGATGATTCCCTGCAG |
| sequence-based reagent | Hs_DOCK4 reverse (*Homo sapiens*) | This paper | qRT-PCR primers | GGAGGTGATGTAACACGACAGG |
| sequence-based reagent | Hs_DOCK5 forward (*Homo sapiens*) | This paper | qRT-PCR primers | GCTTCTGAGCAACATCCTGGAG |
| sequence-based reagent | Hs_DOCK5 reverse (*Homo sapiens*) | This paper | qRT-PCR primers | TCCTTCTCAGCAGCCGTTCCAT |
| sequence-based reagent | Hs_ARL13B forward (*Homo sapiens*) | This paper | qRT-PCR primers | GAACCAGTGGTCTGGCTGAGTT |
| sequence-based reagent | Hs_ARL13B reverse (*Homo sapiens*) | This paper | qRT-PCR primers | GTTTCAGGTGGCAGCCATCACT |
| sequence-based reagent | Hs_DDR2 forward (*Homo sapiens*) | This paper | qRT-PCR primers | AACGAGAGTGCCACCAATGGCT |
| sequence-based reagent | Hs_DDR2 reverse (*Homo sapiens*) | This paper | qRT-PCR primers | ACTCACTGGCTTCAGAGCGGAA |
| sequence-based reagent | Hs_ITGA1 forward (*Homo sapiens*) | This paper | qRT-PCR primers | CCGAAGAGGTACTTGTTGCAGC |
| sequence-based reagent | Hs_ITGA1 reverse (*Homo sapiens*) | This paper | qRT-PCR primers | GGCTTCCGTGAATGCCTCCTTT |
| sequence-based reagent | Hs_RAPGEF2 forward (*Homo sapiens*) | This paper | qRT-PCR primers | CTCGGATCAGTATCTTGCCACAG |

*Appendix 1—key resources table continued*

| Reagent type (species) or resource | Designation | Source or reference | Identifiers | Additional information |
|---|---|---|---|---|
| sequence-based reagent | Hs_RAPGEF2 reverse (*Homo sapiens*) | This paper | qRT-PCR primers | AGGTTCCACTGACAGGCAATGC |
| sequence-based reagent | Hs_RAPGEF6 forward (*Homo sapiens*) | This paper | qRT-PCR primers | AGACAGATGAGGAGAAGTTCCAG |
| sequence-based reagent | Hs_RAPGEF6 reverse (*Homo sapiens*) | This paper | qRT-PCR primers | GACCTCATAGGCACTGGAGACA |
| sequence-based reagent | Hs_APC forward (*Homo sapiens*) | This paper | qRT-PCR primers | AGGCTGCATGAGAGCACTTGTG |
| sequence-based reagent | Hs_APC reverse (*Homo sapiens*) | This paper | qRT-PCR primers | CACACTTCCAACTTCTCGCAACG |
| sequence-based reagent | Mm_PEBP1 forward (*Mus musculus*) | This paper | qRT-PCR primers | ACTCTACACCCTGGTCCTCACA |
| sequence-based reagent | Mm_PEBP1 reverse (*Mus musculus*) | This paper | qRT-PCR primers | TGAGAGGACAGTGCCACTGCTA |
| sequence-based reagent | Mm_BACH1 forward (*Mus musculus*) | This paper | qRT-PCR primers | CCATGACATCCGCAGAAGGAGT |
| sequence-based reagent | Mm_BACH1 reverse (*Mus musculus*) | This paper | qRT-PCR primers | GCGTTGACAGAATGTGGTCTCG |
| sequence-based reagent | Mm_NFATC2 forward (*Mus musculus*) | This paper | qRT-PCR primers | ACTTCACAGCGGAGTCCAAGGT |
| sequence-based reagent | Mm_NFATC2 reverse (*Mus musculus*) | This paper | qRT-PCR primers | GGATGTGCTTGTTCCGATACTCG |
| sequence-based reagent | Mm_ROCK1 forward (*Mus musculus*) | This paper | qRT-PCR primers | CACGCCTAACTGACAAGCACCA |
| sequence-based reagent | Mm_ROCK1 reverse (*Mus musculus*) | This paper | qRT-PCR primers | CAGGTCAACATCTAGCATGGAAC |
| sequence-based reagent | Mm_ROCK2 forward (*Mus musculus*) | This paper | qRT-PCR primers | GTGACCTCAAACAGTCTCAGCAG |
| sequence-based reagent | Mm_ROCK2 reverse (*Mus musculus*) | This paper | qRT-PCR primers | GACAACGCTTCTGAGTTTCCTGC |
| sequence-based reagent | Mm_ADAM10 forward (*Mus musculus*) | This paper | qRT-PCR primers | TGCACCTGTGCCAGCTCTGATG |
| sequence-based reagent | Mm_ADAM10 reverse (*Mus musculus*) | This paper | qRT-PCR primers | GATAGTCCGACCACTGAACTGC |
| sequence-based reagent | Mm_ADAM17 forward (*Mus musculus*) | This paper | qRT-PCR primers | TGTGAGCGGTGACCACGAGAAT |

*Continued on next page*

*Appendix 1—key resources table continued*

| Reagent type (species) or resource | Designation | Source or reference | Identifiers | Additional information |
|---|---|---|---|---|
| sequence-based reagent | Mm_ADAM17 reverse (*Mus musculus*) | This paper | qRT-PCR primers | TTCATCCACCCTGGAG TTGCCA |
| sequence-based reagent | Mm_EPC1 forward (*Mus musculus*) | This paper | qRT-PCR primers | CTGCCAGGCTTCAGTGC TAAAG |
| sequence-based reagent | Mm_EPC1 reverse (*Mus musculus*) | This paper | qRT-PCR primers | ACTGACAGCCTGCTTTCC TACG |
| sequence-based reagent | Mm_PIKFYVE forward (*Mus musculus*) | This paper | qRT-PCR primers | TCTTCTGCCCAGTCCAGCAA TG |
| sequence-based reagent | Mm_PIKFYVE reverse (*Mus musculus*) | This paper | qRT-PCR primers | ACAGAACATGCTCGGACAC TGG |
| sequence-based reagent | Mm_DOCK4 forward (*Mus musculus*) | This paper | qRT-PCR primers | GATAGGAGAGGTGGA TGGCAAG |
| sequence-based reagent | Mm_DOCK4 reverse (*Mus musculus*) | This paper | qRT-PCR primers | CGCCTTGAGATGCAGATCG TAG |
| sequence-based reagent | Mm_DOCK5 forward (*Mus musculus*) | This paper | qRT-PCR primers | GAGCCGACAGTCTCCTCACA TT |
| sequence-based reagent | Mm_DOCK5 reverse (*Mus musculus*) | This paper | qRT-PCR primers | CTGCCTGGTTTTGAAGGTGC TG |
| sequence-based reagent | Mm_ARL13B forward (*Mus musculus*) | This paper | qRT-PCR primers | ACCAGTGGTCTGGCTGAGA TTG |
| sequence-based reagent | Mm_ARL13B reverse (*Mus musculus*) | This paper | qRT-PCR primers | CATCACTGTCCTTC TCCACGGT |
| sequence-based reagent | Mm_DDR2 forward (*Mus musculus*) | This paper | qRT-PCR primers | TCATCCTGTGGAGGCAGTTC TG |
| sequence-based reagent | Mm_DDR2 reverse (*Mus musculus*) | This paper | qRT-PCR primers | CTGTTCACTTGGTGA TGAGGAGC |
| sequence-based reagent | Mm_ITGA1 forward (*Mus musculus*) | This paper | qRT-PCR primers | GGCAGTGGCAAGACCA TAAGGA |
| sequence-based reagent | Mm_ITGA1 reverse (*Mus musculus*) | This paper | qRT-PCR primers | CATCTCTCCGTGGATAGAC TGG |
| sequence-based reagent | Mm_RAPGEF2 forward (*Mus musculus*) | This paper | qRT-PCR primers | GCCGAATGGCATCAG TCAACATG |
| sequence-based reagent | Mm_RAPGEF2 reverse (*Mus musculus*) | This paper | qRT-PCR primers | CAACATCCAGCACTGTGGCG TT |
| sequence-based reagent | Mm_RAPGEF6 forward (*Mus musculus*) | This paper | qRT-PCR primers | ACAGAGTGAGCCAGGTGC TTCA |

*Continued on next page*

*Appendix 1—key resources table continued*

| Reagent type (species) or resource | Designation | Source or reference | Identifiers | Additional information |
|---|---|---|---|---|
| sequence-based reagent | Mm_RAPGEF6 reverse (*Mus musculus*) | This paper | qRT-PCR primers | CACTCACTTCCTCAGTTGGTCC |
| sequence-based reagent | Mm_APC forward (*Mus musculus*) | This paper | qRT-PCR primers | GTGGACTGTGAGATGTATGGGC |
| sequence-based reagent | Mm_APC reverse (*Mus musculus*) | This paper | qRT-PCR primers | CACAAGTGCTCTCATGCAGCCT |
| sequence-based reagent | Hs_HMOX1 forward (*Homo sapiens*) | This paper | qRT-PCR primers (ChIP) | AGTCGCGATTTCCTCATCCC |
| sequence-based reagent | Hs_HMOX1 reverse (*Homo sapiens*) | This paper | qRT-PCR primers (ChIP) | TTCCCTTTGTTTCCGCGAGT |
| sequence-based reagent | Hs_ITGA1 forward (*Homo sapiens*) | This paper | qRT-PCR primers (ChIP) | GGTCTGAGTAACCCCACTTCC |
| sequence-based reagent | Hs_ITGA1 reverse (*Homo sapiens*) | This paper | qRT-PCR primers (ChIP) | AGCACACCACAAAAGCCAAG |
| sequence-based reagent | Hs_DOCK4 forward (*Homo sapiens*) | This paper | qRT-PCR primers (ChIP) | ATTGTTGTGAAGGCCAACCC |
| sequence-based reagent | Hs_DOCK4 reverse (*Homo sapiens*) | This paper | qRT-PCR primers (ChIP) | AGAAGGAGTGCAGTCTGGTTT |
| sequence-based reagent | Hs_RAPGEF2 forward (*Homo sapiens*) | This paper | qRT-PCR primers (ChIP) | GGGTGCTCCAATTGTATGTACTGAT |
| sequence-based reagent | Hs_RAPGEF2 reverse (*Homo sapiens*) | This paper | qRT-PCR primers (ChIP) | TGATTCAGCTTTGGGGAGTGA |
| sequence-based reagent | Hs_PIKFYVE forward (*Homo sapiens*) | This paper | qRT-PCR primers (ChIP) | CTGGACTCCTTCTGCCTGAG |
| sequence-based reagent | Hs_PIKFYVE reverse (*Homo sapiens*) | This paper | qRT-PCR primers (ChIP) | AAGACTCCGCCCTCTGTTTT |
| sequence-based reagent | Hs_ROCK2 forward (*Homo sapiens*) | This paper | qRT-PCR primers (ChIP) | GCATAGGAAGCGAGTACCCAT |
| sequence-based reagent | Hs_ROCK2 reverse (*Homo sapiens*) | This paper | qRT-PCR primers (ChIP) | GACTCCTTTAGGCCCCGTCA |
| sequence-based reagent | Hs_RAPGEF6 forward (*Homo sapiens*) | This paper | qRT-PCR primers (ChIP) | CGCCACAGTTCATTCACACT |
| sequence-based reagent | Hs_RAPGEF6 reverse (*Homo sapiens*) | This paper | qRT-PCR primers (ChIP) | GCGAAGGGTTGTTTGCTAGA |
| sequence-based reagent | Random genomic region, forward (*Homo sapiens*) | This paper | qRT-PCR primers (ChIP) | ATTTGCCTGGAGTGGAAGTG |

*Continued on next page*

*Appendix 1—key resources table continued*

| Reagent type (species) or resource | Designation | Source or reference | Identifiers | Additional information |
|---|---|---|---|---|
| sequence-based reagent | Random genomic region, reverse (*Homo sapiens*) | This paper | qRT-PCR primers (ChIP) | CTGTATCCAGGGGGATGATG |

