## [Decision Letter]

**Acceptance summary:**

The manuscript by Rosner and colleagues describes an interesting approach to therapy for metastasis, which is the predominant cause of death in most solid tumors. This group proposes and tests the intriguing concept that combined, low dose inhibition of multiple kinase signaling networks simultaneously is a more effective strategy overall to inhibit metastasis in models of breast cancer. This work models and establishes a pharmacological means to phenocopy the actions of RKIP, a metastasis suppressor that targets multiple stress kinase signaling networks. Overall, this is a new and interesting idea that has promise and merit. It runs counter to current combination therapy ideas and therefore has promise to be quite impactful.

**Decision letter after peer review:**

Thank you for submitting your article "Limited inhibition of multiple nodes in a driver network blocks metastasis" for consideration by *eLife*. Your article has been reviewed by 3 peer reviewers, and the evaluation has been overseen by Maureen Murphy as the Senior and Reviewing Editor. The following individual involved in review of your submission has agreed to reveal their identity: Rebecca Riggins (Reviewer #1).

The reviewers have discussed the reviews with one another and the Reviewing Editor has drafted this decision to help you prepare a revised submission.

Summary:

The manuscript by Rosner and colleagues describes an interesting approach to therapy for metastasis, which is the predominant cause of death in most solid tumors. This group proposes and tests the intriguing concept that combined, low dose inhibition of multiple kinase signaling networks simultaneously is a more effective strategy overall to inhibit metastasis in models of breast cancer. This work models and establishes a pharmacological means to phenocopy the actions of RKIP, a metastasis suppressor that targets multiple stress kinase signaling networks. Overall, this is a new and interesting idea that has promise and merit. It runs counter to current combination therapy ideas and should be evaluated more broadly. There is potential for this interesting contribution to be quite impactful, provided that major concerns are addressed.

Essential revisions:

1. A key issue brought up by all three reviewers is with the number and types of replicates of the experiments. In particular, many of the experiments appear to be performed with technical replicates, e.g., measuring the same sample three times, rather than biological replicates e.g. with different cells on different days. Technical replicates are not representative of biological variation, so this is not acceptable practice. A few examples:

a. Figure 2F. Representative of 2 independent experiments? A third replicate needs to be performed, and quantitative data for all three, with error bars, should be shown.

b. Figure 2G. No error bars – apparently each experiment was performed once. This is not considered acceptable.

c. Figure 3D: Data represents 3 technical replicates? These need to be biological replicates, and preferably performed on different days, and on different plates.

d. Figure 5 E-G. "n=3 technical replicates". Running the same sample 3 times is not acceptable. Biological replicates (wherein different batches of cells are treated and then measured independently) are required.

e. Figure 5 H-J: 'bar graphs are representative of 2 independent experiments' – again, this is not acceptable.

f. Supp. 5F – Technical replicates are not sufficient. Biological replicates (at least n=3) need to be performed. Also why are different concentrations shown for different lines? Were other concentrations tested? If so, this data should be shown.

Additional details in the main and supplementary figure legends would be helpful to readers. These include consistent reporting of the number of biological and technical replicates for every figure.

2. A second major issue is more conceptual. While experimentally this is a very comprehensive and elegant study, "prevention" or "blocking" of metastasis has key fundamental downfalls as a therapeutic strategy. Metastatic sites are very often already seeded by the time the primary breast tumor is diagnosed (e.g. PubMed IDs 27931250 and 28810143). Recent modeling suggests that this can occur as much as four (4) *years* prior to detection of the primary lesion (PubMed IDs 31209394 and 32424352). A 4-drug regimen selected for its capacity to block invasion, but not impair proliferation, is therefore unlikely to be a viable strategy in this context. As a proof-of-concept for how multi-kinase inhibition can be achieved without compensatory activation of parallel networks, this study is outstanding. As a roadmap for how to meaningfully address metastatic disease as it presents clinically, this study is limited. Because the authors take such a strong stance on the promise of their approach, I would like to see a more thorough and thoughtful discussion of this major limitation in a revised manuscript.

3. A notable achievement of this study is identification of a 4-drug regimen that inhibits stress MAPK signaling without driving a compensatory increase in PI3K activation, as measured by Akt phosphorylation in in vitro studies (Figure 6D, Supplementary Figure 7D). Does this also translate to the in vivo setting, i.e. in the lung metastases shown in Figure 4E or the primary tumors shown in Figure 4I, are pAkt levels unchanged or even moderately reduced? Inclusion of these data would markedly strengthen one of the key conclusions of the work.

4. Lines 187-207. The goal of these experiments appears to be to show that the 4D combination affects metastasis. Yet at both of the concentrations tested there is a significant impact on tumor proliferation, thus confounding the analysis of metastatic burden. By only using 2 concentrations, both of which have a strong effect on proliferation, the authors have failed to show that this is a 'metastasis effect' versus a tumor cell survival and proliferation effect.

5. There is some in vivo data for single agent MEK inhibitors, but it seems it was not reported how such single agent treatments inhibit metastasis compared to the 4D regimen. This would be important to establish the increased efficacy of the 4D treatment compared to single agents-the central premise of the paper.

---

## [Author Response]

Essential revisions:1. A key issue brought up by all three reviewers is with the number and types of replicates of the experiments. In particular, many of the experiments appear to be performed with technical replicates, e.g., measuring the same sample three times, rather than biological replicates e.g. with different cells on different days. Technical replicates are not representative of biological variation, so this is not acceptable practice. A few examples:a. Figure 2F. Representative of 2 independent experiments? A third replicate needs to be performed, and quantitative data for all three, with error bars, should be shown.b. Figure 2G. No error bars – apparently each experiment was performed once. This is not considered acceptable.c. Figure 3D: Data represents 3 technical replicates? These need to be biological replicates, and preferably performed on different days, and on different plates.d. Figure 5 E-G. "n=3 technical replicates". Running the same sample 3 times is not acceptable. Biological replicates (wherein different batches of cells are treated and then measured independently) are required.e. Figure 5 H-J: 'bar graphs are representative of 2 independent experiments' – again, this is not acceptable.f. Supp. 5F – Technical replicates are not sufficient. Biological replicates (at least n=3) need to be performed. Also why are different concentrations shown for different lines? Were other concentrations tested? If so, this data should be shown.Additional details in the main and supplementary figure legends would be helpful to readers. These include consistent reporting of the number of biological and technical replicates for every figure.

We have now ensured that all figures included in the manuscript represent at least 3 biological replicates. Thus, we have repeated all key experiments in the entire paper to make sure there are at least 3 independent replicates in at least two different cell lines. This information is now in the figure legends, and graphs are plotted to include the results of at least 3 independent experiments with appropriate statistics.

Specific figures that have been updated and other changes in manuscript figures are as follows:

1. We updated the ChIP data in Figure 1E with more targets and added ChIP-seq data supporting our findings in the revised Figure 1—figure supplement 2C.

2. We removed Figure 2F and Supplementary Figure 3A from the original manuscript, an *in vitro* validation of the mass spectrometry tumor analysis, since this technique has been extensively validated ^1-4^, *in vitro* kinase activities with RKIP are not as robust as in tumors, and an *in vitro* analysis of RKIP effects on multiple kinases was just an additional control that is not relevant to the multi-drug focus of the paper.

3. We limited the content of the original Figure 2G to the specific responses to the MAPK inhibitors that differ between the three TNBC cell lines and moved this figure to Figure 5—figure supplement 1A-D.

4. We added individual data points and error bars to the graph in Figure 3B as requested by the reviewers.

5. We repeated the invasion and proliferation assays originally in Figure 3D,E to represent three independent experiments in each cell line. We also moved the data from additional cell lines from the Supplementary to the main figure (now Figure 3D-F)

6. We updated the network topology used in mathematical modeling in Figure 5A-D and Figure 6A-C with interactions that we validated experimentally in at least three independent experiments. We removed only one interaction (node 6 *JNK* – node 8 p38) from the network and re-performed the modeling analysis. The conclusions from the modeling remained the same.

7. We repeated all the BACH1 qRT-PCR analyses depicted in Figure E-J in at least three independent experiments and updated the graphs accordingly. The conclusions from the experiments remained unchanged.

8. For the original Supplementary Figure 1D, we replaced *in vitro* data with *in vivo* data from xenograft tumors expressing RKIP as it is more relevant to the study (now Figure 1—figure supplement 1D)

9. For the original Supplementary Figure 2C and D, we substituted *in vivo* data using BACH1-depleted tumors for *in vitro* data as it is more relevant to the study (now Figure 1—figure supplement 2D)

10. We removed invasion assays for MLK, TAOKs and other inhibitors (originally Supplementary Figure 3B-D) and just put in references to the literature instead since these findings have already been published previously.

11. We have added the results from each independent experiment that led us to the BM1 MAPK network topology (currently Figure 2F) as a separate figure supplement (Figure 2—figure supplement 1B). In the same figure, we also added dose-response studies demonstrating downregulation of phosphorylation on direct substrates of p38, *JNK*, and MEK when these kinases are blocked by the specific inhibitors used in this study. (Figure 2—figure supplement 1A)

12. We divided the original Supplementary Figure 5 into two figures. The new Figure 3—figure supplement 1 shows the results from individual high-throughput invasion assays that lead us to 4D-MAPKi as a candidate treatment. The new Figure 3—figure supplement 2 summarizes the updated results from three independent experiments in three different cell lines originally depicted in Supplementary Figure 5E.

13. We dedicated the new Figure 5—figure supplement 1A-D to clearly demonstrating the differences in network topology among different cell lines or even under different stimuli (each with at least three independent experiments). These figures and graphs show the differences with respect to the reference BM1 network topology (Figure 2F, Figure 2—figure supplement 1)

14. We have added a summary graph from three independent experiments showing BACH1 protein downregulation by 4D-MAPKi in Figure 5—figure supplement 1E (originally Supplementary Figure 7B).

15. We added a new mouse experiment comparing lung metastases in the syngeneic LMB mouse model under 4D-MAPKi regimen vs. the individual drugs, as requested by the reviewers (Figure 5—figure supplement 1F).

16. We added new p-AKT western blot analysis from syngeneic LMB tumors treated with 4D-MAPKi to show that 4D-MAPKi treatment does not activate the compensatory AKT signaling *in vivo* as requested by the reviewers (Figure 6—figure supplement 1C).

17. Author list, Methods, Acknowledgments, Author Contributions and Declaration of Interest sections have been updated with the relevant information.

2. A second major issue is more conceptual. While experimentally this is a very comprehensive and elegant study, "prevention" or "blocking" of metastasis has key fundamental downfalls as a therapeutic strategy. Metastatic sites are very often already seeded by the time the primary breast tumor is diagnosed (e.g. PubMed IDs 27931250 and 28810143). Recent modeling suggests that this can occur as much as four (4) years prior to detection of the primary lesion (PubMed IDs 31209394 and 32424352). A 4-drug regimen selected for its capacity to block invasion, but not impair proliferation, is therefore unlikely to be a viable strategy in this context. As a proof-of-concept for how multi-kinase inhibition can be achieved without compensatory activation of parallel networks, this study is outstanding. As a roadmap for how to meaningfully address metastatic disease as it presents clinically, this study is limited. Because the authors take such a strong stance on the promise of their approach, I would like to see a more thorough and thoughtful discussion of this major limitation in a revised manuscript.

We appreciate the point raised by this reviewer and agree that it is important and requires clarification. We have now discussed this issue more extensively in the Discussion section of the paper. Briefly, our point is that, in addition to using cytotoxic agents such as radio-, chemo- or immunotherapy to kill the main tumor, one must address metastatic progression since this is a dynamic and highly drug-resistant process. While early metastatic seeding can take place before the primary tumor is clinically detectable, the primary tumor continuously sheds metastatic cells into the circulation that can form metastases at other sites^5, 6,7^. Therefore, an anti-metastatic treatment that slows down this dynamic spread would have therapeutic benefit even at the late stage of disease. In addition, evidence also suggests that metastatic cells are more resistant to systemic treatments due to their more mesenchymal phenotype, and low proliferation rates^8^. If an anti-metastatic therapy such as 4D-MAPKi can revert metastatic cancer cells back to a less metastatic, more epithelial-like state, it can sensitize these cells to certain systemic and metabolic treatments as we have previously shown^9^. Therefore, we are suggesting a 2-part strategy to convert metastatic cells to a non-metastatic state prior to treatment with agents that will suppress proliferating cells.

3. A notable achievement of this study is identification of a 4-drug regimen that inhibits stress MAPK signaling without driving a compensatory increase in PI3K activation, as measured by Akt phosphorylation in in vitro studies (Figure 6D, Supplementary Figure 7D). Does this also translate to the in vivo setting, i.e. in the lung metastases shown in Figure 4E or the primary tumors shown in Figure 4I, are pAkt levels unchanged or even moderately reduced? Inclusion of these data would markedly strengthen one of the key conclusions of the work.

We agree and have now included this experiment. We checked the activation of p-AKT(Ser473) in the syngeneic LMB primary tumors treated with either the vehicle control or the 4D-MAPKi regimen. In this study, we observed no significant activation of the compensatory p-AKT signaling in the primary tumors treated with the 4D-MAPKi regimen, consistent with our predictions (see Figure 6—figure supplement 1C in the revised manuscript).

4. Lines 187-207. The goal of these experiments appears to be to show that the 4D combination affects metastasis. Yet at both of the concentrations tested there is a significant impact on tumor proliferation, thus confounding the analysis of metastatic burden. By only using 2 concentrations, both of which have a strong effect on proliferation, the authors have failed to show that this is a 'metastasis effect' versus a tumor cell survival and proliferation effect.

We thank the reviewer for raising this important point and appreciate the opportunity to clarify our conclusions. We used two different models to illustrate the effect on metastasis. When using the spontaneous metastasis model where metastasis is derived from the primary tumor, as noted by the reviewer, it is difficult to distinguish inhibitory effects on metastasis alone from inhibitory effects on primary tumor growth. (All cells that grow at either primary or metastatic sites have a survival component, so we are assuming here that survival at the metastatic site is an aspect of metastasis.) We successfully distinguished growth from invasion *in vitro* and, using a similar strategy, identified drug concentrations that individually did not inhibit growth in vivo. However, since drug effects are usually not additive, it is challenging to precisely titrate the 4 drugs so that we suppress metastasis but don’t affect primary tumor growth *in vivo.* Therefore, as an alternative assay that we and others have used in the past to monitor metastasis alone (e.g. ^15^), we utilized the tail-vein or cardiac injection model that focuses on extravasation and metastatic colonization using comparable numbers of tumor cells. These assays clearly show a robust inhibition by the 4-drug combination when the same number of cells are present in the circulation. Even a limited 2-day treatment with drug after tail vein injection suppressed metastatic growth, suggesting that 4D-MAPKi primarily affects the early steps of metastatic seeding such as extravasation, invasion, or colonization. Ultimately, as noted above, our goal is to suppress metastasis as well as growth of the primary tumor. As the reviewer points out, this 4-drug combination has the potential to do both. This point will now also be included in the paper.

5. There is some in vivo data for single agent MEK inhibitors, but it seems it was not reported how such single agent treatments inhibit metastasis compared to the 4D regimen. This would be important to establish the increased efficacy of the 4D treatment compared to single agents-the central premise of the paper.

We appreciate the potential concern as well as the opportunity to clarify the premise of our study. We realize that, in certain tumor models, single agents such as the MEK inhibitor suppress metastasis. Our studies suggest, however, that the 4D regimen is more likely to have efficacy across different cellular networks and different environmental conditions and therefore work better when treating heterogeneous tumors. We have now done the experiment requested by the reviewer to compare *in vivo* efficacy for single agent MEK or the other inhibitors (p38, *JNK*, MLK) versus the 4D regimen using the syngeneic LMB mouse model. When we tested the drugs on LMB cells in culture, the MEK and *JNK* inhibitors and the 4D combination were effective at suppressing MAPK network output (assessed by the pro-metastatic gene BACH1) whereas the other inhibitors were not. Similar to these cell culture studies, both MEK and 4D drug regimens reduced metastatic burden in the lungs of the mice but p38 inhibitors did not (Figure 5—figure supplement 1F). The fact that *JNK* inhibitors inhibited BACH1 expression *in vitro* but did not suppress metastasis *in vivo* maybe due to a number of factors including drug accessibility, different outputs, or microenvironmental interactions. These results are consistent with our original findings and still support the rationale for utilizing multi-drug combinations.

In addition, it should be noted that, while MEK inhibitors as single agents can be very effective in blocking tumor growth and metastasis in preclinical settings, they rarely work long term in clinical settings because of the compensatory activation of resistance pathways such as PI3K/AKT signaling. Our studies suggest that 4D-MAPKi may exhibit similar or better efficacy to MEK inhibitors without activating these compensatory pathways (as discussed in Essential Revisions 3 above). These points are now discussed in the Discussion section of the paper.References:

1. Duncan, J.S. et al. Dynamic reprogramming of the kinome in response to targeted MEK inhibition in triple-negative breast cancer. Cell 149, 307-321 (2012).2. Zawistowski, J.S. et al. Enhancer Remodeling during Adaptive Bypass to MEK Inhibition Is Attenuated by Pharmacologic Targeting of the P-TEFb Complex. Cancer Discov 7, 302-321 (2017).3. Stuhlmiller, T.J. et al. Inhibition of Lapatinib-Induced Kinome Reprogramming in ERBB2-Positive Breast Cancer by Targeting BET Family Bromodomains. Cell reports 11, 390-404 (2015).4. Cooper, M.J. et al. Application of multiplexed kinase inhibitor beads to study kinome adaptations in drug-resistant leukemia. PLoS One 8, e66755 (2013).5. Quinn, J.J. et al. Single-cell lineages reveal the rates, routes, and drivers of metastasis in cancer xenografts. Science 371 (2021).6. Kim, M.Y. et al. Tumor self-seeding by circulating cancer cells. Cell 139, 1315-1326 (2009).7. Gupta, G.P. and Massague, J. Cancer metastasis: building a framework. Cell 127, 679-695 (2006).8. The "something more" than interpretation revisited: sloppiness and co-creativity in the psychoanalytic encounter. J Am Psychoanal Assoc 53, 693-729; discussion 761-699 (2005).9. Lee, J. et al. Effective breast cancer combination therapy targeting BACH1 and mitochondrial metabolism. Nature 568, 254-258 (2019).10. Heinrich, R., Neel, B. and Rapoport, T. Mathematical models of protein kinase signal transduction. Molecular cell 9, 957-970 (2002).11. Beguerisse-Diaz, M., Desikan, R. and Barahona, M. Linear models of activation cascades: analytical solutions and coarse-graining of delayed signal transduction. J R Soc Interface 13 (2016).12. Heinrich, R. and Rapoport, T.A. A linear steady-state treatment of enzymatic chains. General properties, control and effector strength. Eur J Biochem 42, 89-95 (1974).13. Kacser, H. and Burns, J.A. The control of flux. Biochem Soc Trans 23, 341-366 (1995).14. Chaves, M., Sontag, E.D. and Dinerstein, R.J. Optimal length and signal amplification in weakly activated signal transduction cascades. J. Phys. Chem. B 108, 15311–15315 (2004).15. Huang, C. and Ferrell, J. Ultrasensitivity in the mitogen-activated protein kinase cascade. Proceedings of the National Academy of Sciences of the United States of America 93, 10078-10083 (1996).16. Brandman, O., Ferrell, J.E., Jr., Li, R. and Meyer, T. Interlinked fast and slow positive feedback loops drive reliable cell decisions. Science 310, 496-498 (2005).17. Friedmann, E. PDE/ODE modeling and simulation to determine the role of diffusion in long-term and -range cellular signaling. BMC Biophys 8, 10 (2015).